# The momentum of the solar energy transition

Femke J. M. M. Nijsse [1] ✉, Jean-Francois Mercure [1,2,3], Nadia Ameli [4], Francesca Larosa [4,5], Sumit Kothari [4], Jamie Rickman[4], Pim Vercoulen [1,6] & Hector Pollitt[2,3]

Decarbonisation plans across the globe require zero-carbon energy sources to be widely deployed by 2050 or 2060. Solar energy is the most widely available energy resource on Earth, and its economic attractiveness is improving fast in a cycle of increasing investments. Here we use data-driven conditional technology and economic forecasting modelling to establish which zero carbon power sources could become dominant worldwide. We find that, due to technological trajectories set in motion by past policy, a global irreversible solar tipping point may have passed where solar energy gradually comes to dominate global electricity markets, without any further climate policies. Uncertainties arise, however, over grid stability in a renewables-dominated power system, the availability of sufficient finance in underdeveloped economies, the capacity of supply chains and political resistance from regions that lose employment. Policies resolving these barriers may be more effective than price instruments to accelerate the transition to clean energy.

A rapid transformation of the energy system is necessary to keep warming well below 2 °C, as set out in the Paris Agreement and reinforced in the Glasgow Pact. Many countries have committed to achieving net-zero targets by 2050 (incl. EU, UK, Japan, Korea), 2060 (China) or 2070 (India). Net-zero targets imply mass-scale deployment of zero-carbon energy technologies such as solar and wind power, likely in combination with negative emission technologies[1]. However, the potential for negative emissions to compensate positive emissions remains relatively limited[2,3].

Renewables have historically been considered expensive, their deployment requiring high subsidies or carbon taxes[4,5]. However, following a fruitful history of innovation and past climate policy, renewables now increasingly compete with fossil fuels[6,7]. Whether renewables become the new normal increasingly hinges upon industry and trade development rather than a pure normative necessity to meet carbon budgets[7–9]. Policy-makers urgently need to know not only whether a renewables future is possible, but whether it is materialising.

Between 2010 and 2020, the cost of solar PV fell by 15% each year, representing a technological learning rate of around 20% per doubling of installed capacity[8]. At the same time, the installed capacity has risen by 25% per year, causing and partly caused by these cost reductions. Meanwhile, onshore wind capacity grew by 12% a year, with a learning rate of 10% per doubling of capacity[8,9]. If these rates of rapid co-evolution are maintained, solar PV and wind power appear ready to irreversibly become the dominant electricity technologies within 1-2 decades, as their costs and rate of growth far undercut all alternatives. Were that to be the case, a renewables tipping point in the power sector could be imminent or even already have been passed, and the policy and finance spheres should prepare for a rapid disruptive transition. Despite this evidence, the energy modelling community has not yet identified this possibility with any degree of consensus[8], suggesting instead that fossil fuel-dominated electricity systems would likely continue as a result from inadequate carbon pricing.

The problem of high cost for renewables has changed into a problem of balancing electricity grids, in which large amounts of

---

[1]Global Systems Institute, Department of Geography, University of Exeter, Exeter, UK. [2]Cambridge Centre for Energy, Environment and Natural Resource Governance, University of Cambridge, Cambridge, UK. [3]The World Bank, Washington, DC, USA. [4]Institute for Sustainable Resources, University College London, London, UK. [5]Royal Institute of Technology (KTH), Climate Action Centre, Stockholm, Sweden. [6]Cambridge Econometrics, Cambridge, UK. ✉ e-mail: f.j.m.m.nijsse@exeter.ac.uk

intermittent wind and solar generation pose challenges. Batteries play an important role in mitigating that issue and show a similarly high learning rate[10]. This implies that electricity storage costs and diffusion could follow a comparable and coupled trajectory to PV in the 2020s.

Whether solar and wind can dominate electricity grids depends on the ability of the technology to overcome a series of barriers. This includes how to deal with the seasonal variation for which batteries are ill-suited[11]. The cost of managing large amounts of intermittency could offset further cost reductions in solar panels and wind turbines, impeding their rapid diffusion[12]. The unequal availability of finance to support solar and wind investments in various countries[13] may be an issue, too. Supply chains may be poorly prepared for such a rapid technological roll out[14]. Finally, political resistance in areas of declining fossil fuel use or trade could curb the willingness of governments to embrace a solar revolution[15].

Here, we use a global, data-driven energy-technology-economy simulation model (E3ME-FTT) to conditionally forecast the deployment of energy technologies up to 2060, under current policy regimes. We focus on identifying the existence of a tipping point for solar and wind, assuming that no further policy is adopted to usher in a solar and wind-dominated electricity system. We then explore in detail the various barriers that could impede this renewables revolution, and identify what non-traditional policies could be used to bridge those gaps.

Historical projections of energy generation have consistently underestimated uptake rates of solar energy[16,17]. For example, only a year after the publication of the 2020 World Energy Outlook (WEO), the IEA's "Stated policies scenario" has been revised strongly in favour of solar energy. Nevertheless, the total share of solar in power production only reaches 20% by 2050 in that baseline scenario despite historically low prices[18]. Systematic underestimation of low-carbon technology deployment in energy models could stem from systematic lack of suitable or realistic representation of induced innovation and diffusion processes[19–21].

Solar energy started its journey in niche markets, like most innovations, supplying electricity to applications where little alternatives existed in space and remote locations[22]. Since then, cumulative investments and sales, driven by past policy, have made its cost come down by almost three orders of magnitude. The introduction of feed-in tariffs in mainly Germany induced a volume of investment and related cost reductions, that brought the technology to mainstream markets following Chinese involvement in supply chains[7].

Cost reductions and rapid deployment work hand in hand, something observed for many technologies[7]. Deployments typically follow Rogers' S-curve diffusion[23], with a bi-directional interaction with cost reductions from Wright's law[24]. For solar (and wind), rapid deployments, supported by past policies, have pushed down technology costs. This promotes further diffusion in a virtuous cycle[7]. Such nonlinearity in the diffusion process raises the possibility of an irreversible tipping point[25].

There are many reasons why solar has experienced such high learning rates. Its simplicity, modularity and mass-scale replicability allow for significant learning opportunities, related to those seen across the electronics industry[26–28]. Indeed, numerous spillovers have originated from the computer industry[22]. Innovation and improvements to solar PV are ongoing. For instance, the commercialisation of (hybrid) perovskite cells holds promises for higher efficiencies and lower unit prices[29,30]. Due to decreasing technology risks and financial learning, finance is partly cheaper to procure[31]. Progress in recycling helps material supply security and may decrease life-cycle costs[32]. Meanwhile, the chemical diversity of batteries, a storage technology highly supportive of solar PV, makes it likely that further cost declines can be achieved[33].

The historical failure of the modelling community to anticipate the rapid progress of solar power could stem from an over-reliance on outdated data, the lack of use of learning curves, and the imposition of maximum deployment levels and floor costs[16,34]. As the primary innovation in this paper, forecasting technology evolution and induced innovation can more effectively be achieved based on evolutionary simulations, using the most recent data available, that focus on the two-way positive feedback between induced innovation and diffusion[24,35].

This work supplements recent research by Way et al.[16]. Way et al. developed a probabilistic empirically validated global model of energy technology costs. The research showed that a scenario of high renewables uptake leads to a significantly lower-cost energy system, and the authors argued that energy models should be updated to reflect the high probability of low-cost renewables. This paper differs in two key ways: 1) we do not impose a scenario, but rather allow investor decisions to dictate the deployment of technologies. 2) we use a globally disaggregated model and look at region-specific alternative sources of electricity.

## Results

### Towards a new baseline scenario

Following the recent progress of renewables, fossil fuel-dominated projection baselines are not realistic anymore. Here, we focus on the co-evolving dynamics of diffusion and innovation to project the mid to long-term diffusion trajectory of 24 power technologies. We use the historical data-driven E3ME-FTT integrated energy-economy model, in which a system dynamics simulation method, combined with choice modelling (see Methods), tracks the positive feedbacks that emerge between cost reductions and diffusion, something not usually represented in models that have fixed yearly learning[5]. We use IEA data for historical generation, CAPEX and OPEX, BNEF for capacity factors, construction and lifetimes until 2020, IRENA for historical renewables capacity data between 2019 and 2021.

Technological trajectories typically have inertia in their diffusion that depend on their lifecycle turnover, with half-lives ranging between 10 and 15 years for short-lived units (cars), 25–40 years for fossil fuel plants, and 50–100 years for long-lived infrastructure, such as nuclear plants and hydro dams[36]. These long lifetimes prevent technological trajectories from changing direction abruptly. This autocorrelation time in the direction of evolution (or degree of inertia) implies that energy system technological forecasting constrained by observed diffusion and cost trajectories, as done here, can be reliable within at least 15-20 years, subject to an increasing error that cumulates over the simulation time span.

Figure 1 shows the global share of electricity production of 11 key technologies (Supplementary Figure 1 for a regional breakdown). The current mix is highly varied. By mid-century, according to E3ME-FTT, solar PV will have come to dominate the mix, even without any additional policies supporting renewables. This is due to solar costs declining far below the costs of all alternatives, while its parent industrial supply capacity increases rapidly. Its scale expands, because of its current rapid and exponential diffusion trajectory and comparatively high learning rate. Even the market shares of onshore and offshore wind power in the global electricity mix start declining around 2030, outpaced by solar. This is due to a lower learning rate of wind compared to solar and a growing cost gap in the model. However, onshore continues growing in absolute terms until 2040, and offshore to the end of the simulation. Concentrated solar power grows over the entire period, but without targeted policy its overall share in the power mix remains small, despite its advantage as a dispatchable source of electricity.

The trend towards renewables dominance (Fig. 2a) and notably solar PV (Fig. 2b) appears imminent in China, and lags in Africa and Russia. Africa lags despite a very high technical potential and low seasonality. The slow uptake can mostly be attributed to nonpecuniary aspects (grid flexibility, trust in new technologies), which requires

prices to fall further below alternatives before there is significant uptake. This occurs after uptake by other countries drives down prices further.

The levelised cost of electricity (LCOE$_{ssc}$, which includes system storage costs, see Methods) is shown in Fig. 3. We tentatively assign additional system costs for storage to be borne by renewable energy producers. Even though storage needs increase substantially over time, LCOE for solar energy decreases overall. This is because the learning rate for short-term storage is very high, and the learning rate for long-duration storage (we assume hydrogen is used for seasonal storage) is expected to be relatively high too[8]. Of the major countries shown, solar PV is initially more expensive than coal only in Japan, where cost-parity is reached around 2025.

In 2020, wind energy has the lowest LCOE in a majority the 70 regions defined in the E3ME-FTT models (Fig. 4). Where this is not the case, solar PV, nuclear or coal dominate. By 2030, this has flipped, in favour in solar power across most of the world (see Supplementary Figs. 2 and 3 for worst/best case maps). We assume a uniform declining cost per kW of PV panels worldwide, with differing solar irradiation for each region. This assumption is based on empirical findings[37]. Due to

this international spillover effect, most regions of the world are likely going to gain access to low-cost solar energy. As such, a region may reach cost parity between solar and the cheapest alternative through the influence of other countries on the scale of production and costs, even if cumulative investments in that region are modest. This implies that developing countries could become realistic markets for solar energy even when the capacity of their governments to implement climate policies remains limited.

Figure 5 shows the robustness of the result to a set of model assumptions (see Methods). The two most important sources of uncertainty are potential delays in making necessary grid adjustments and the learning rate for wind power. If installing solar power plants takes twice as long due to delays with grid expansions, the median share of solar in 2050 drops by 16 percentage points. Notably, with solar prices far below alternatives, higher learning rates have a small effect on diffusion. Overall, in 72% of the simulations done for robustness testing, solar makes up more than 50% of power generation in 2050. This suggests that solar dominance is not only possible but also likely.

These projections and sensitivities give us some confidence to suggest that realistic energy model baselines should, from now on, include substantially larger shares of solar energy than what is commonly assumed, as they make coal and gas-dominated baseline scenarios largely unrealistic. The main scenario framework assessed in the IPCC reports, the socialeconomic pathways (SSPs), include scenarios with increasing reliance on coal to the energy mix[38]. This work notably indicates these scenarios are highly improbable.

The above projections appear robust with respect to cost and technical factors included in the model. However, systemic problems not modelled could, nevertheless, develop into barriers hindering achieving climate targets. This suggests that further climate policy action should focus on addressing these barriers.

## Overcoming barriers
We highlight four barriers that go beyond considerations of levelized costs and a) may significantly slow down the solar tipping point if unaddressed b) are global and c) are not fully implemented into integrated assessment models. The four identified encompass the technological, policy, market and economic, regulatory, political and social barriers identified by the literature[39] as the most relevant for solar PV deployment in the next three decades.

As a first barrier, we consider grid resilience. In many published energy scenarios with higher shares of solar and wind power, "dark doldrums", periods of simultaneously low wind speeds and solar irradiation, form the predominant vulnerability[40]. From geophysical constraints, it is possible to compute an optimal mix of wind and solar power, which maximises the match between supply and demand. The

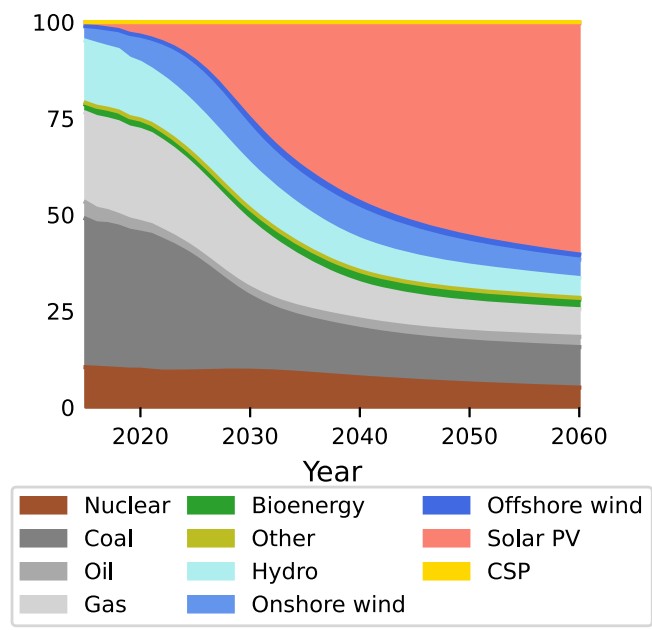

**Fig. 1 | Worldwide share in electricity production of various technologies.** In 2020, fossil fuels produce 62% of electricity. This percentage reduces to 21% in 2050, with solar responsible for 56% of production.

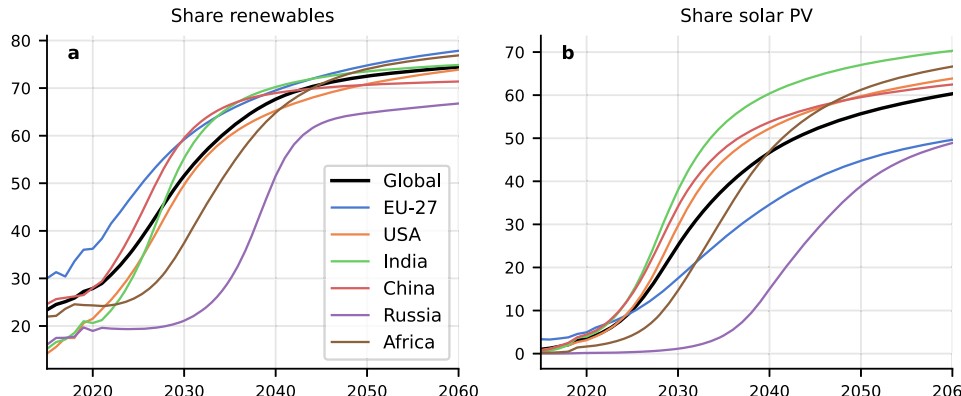

**Fig. 2 | Renewable share of electricity. a** total renewables (hydro + wind + solar + biomass) and (**b**) solar PV. Initially, renewables are dominated by hydropower and to a lesser extent wind. This is soon overtaken by solar, depending on regional factors.

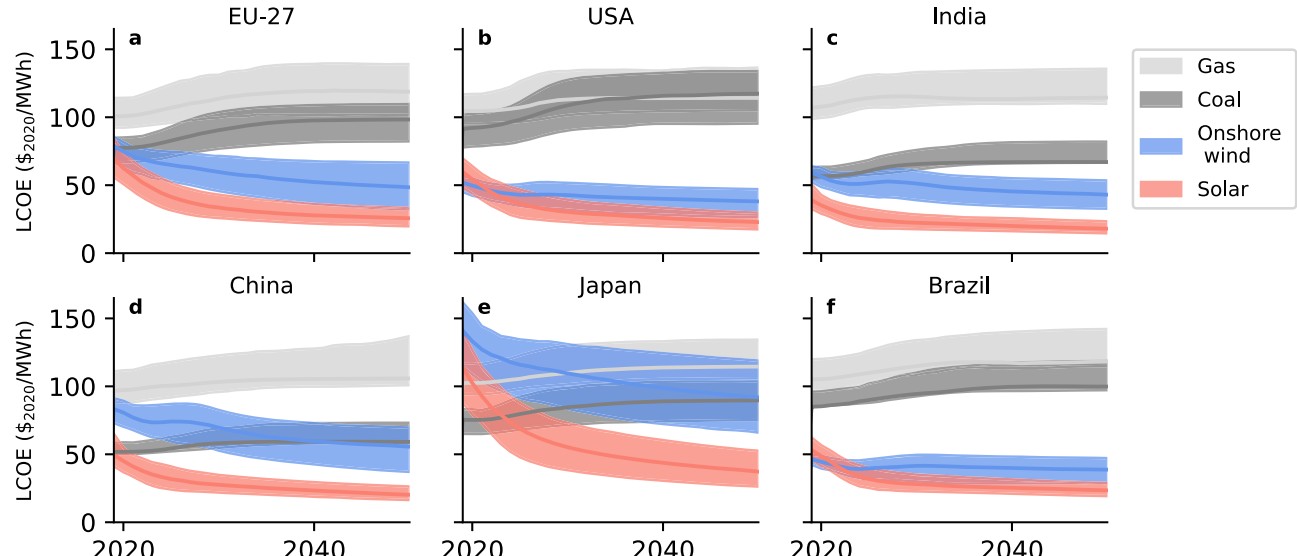

**Fig. 3 | Regionally weighted average levelised cost of electricity (LCOE), including system storage costs and excluding policies. a** EU-27 (**b**) United States (**c**) India (**d**) China (**e**) Japan and (**f**) Brazil. Shaded areas are the 10–90% confidence interval. Solar PV + system storage is already among the cheapest forms of electricity. In some regions, wind and solar remain competitive, whereas solar becomes much cheaper in others. Without carbon taxation, coal is typically cheaper than gas.

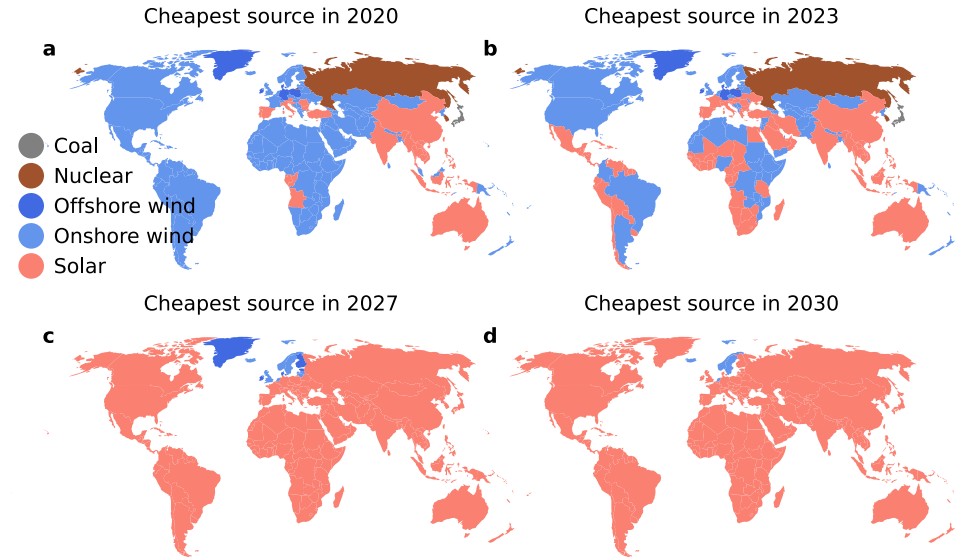

**Fig. 4 | Technology with the lowest LCOE$_{ssc}$ by year and E3ME region.** Each map shows the 70 E3ME regions: in 2020 (**a**), 2023 (**b**), 2027 (**c**) and 2030 (**d**). The biggest shift occurs between 2020 and 2027, which sees a range of technologies give way to solar PV as the cheapest source of electricity.

typical optimal share of solar when 12 h of battery storage is available lies between 10–70%, depending on geography. Where less storage is available, the optimal mix shifts towards more wind power[11]. When either of the two main technologies is (near)-absent, the grid becomes more vulnerable to weather fluctuations. As such, solar-dominated grids may not be desirable. Importantly, no mechanism guarantees that optimal grids are achieved if left to market forces, especially in contexts of diverging technology costs, and solar dominance could become self-limiting. While E3ME-FTT models grid constraints of a typical year, weather extremes are not considered.

The self-limiting effect of solar PV diffusion due to intermittency can be overcome with a policy mix supporting wind power and other zero-carbon energy sources, as well as improved storage, grid connections and demand-response. Notably, new power market rules can be designed to incentivise investment in generators that complement

solar production on a daily to seasonal scale, according to the savings in storage that they generate. Specifically, our model suggests that the allocation of storage costs to the grid and charged directly to consumers incentivises more renewables diffusion than requiring renewables to carry the full burden of storage needs (see Fig. 5), leading to lower overall system costs[41].

Secondly, the availability of finance may act as a barrier. Solar growth trajectories will inevitably depend on the availability of finance. Low-carbon finance is presently highly concentrated in high-income countries[42]. Even international North-South flows largely favour middle-income countries, leaving lower income countries – particularly those in Africa – deficient in solar finance despite the enormous investment potential[42].

This unequal distribution of finance reflects different investment risk considerations across countries. Differences in local financial

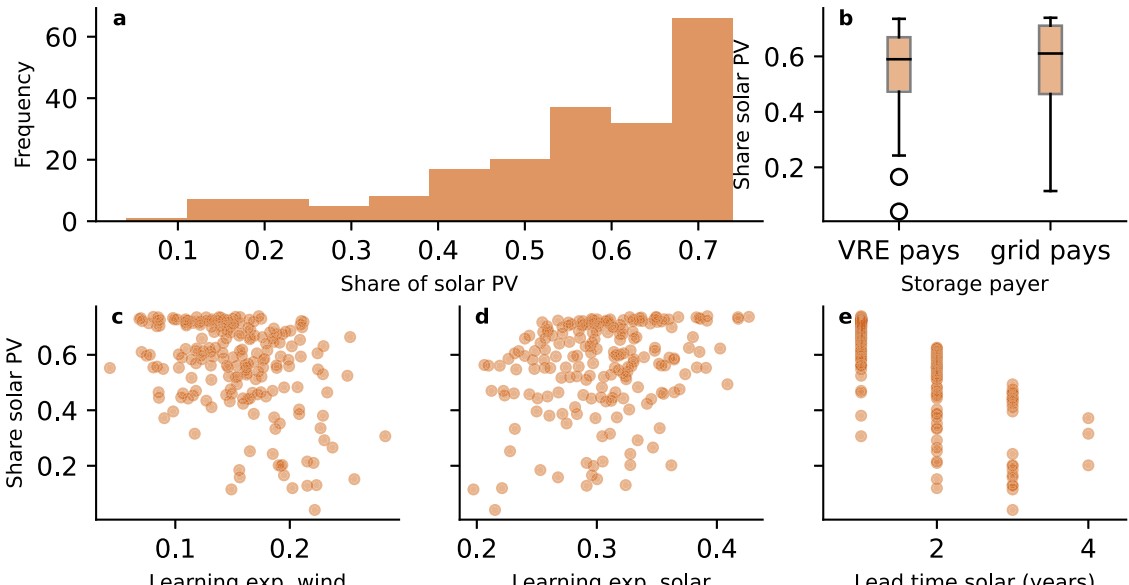

**Fig. 5 | Shares of solar PV in the power sector when varying key inputs. a** The overall histogram of the 2050 shares of solar PV. **b** The shares solar PV depending on who pays for storage costs (variable renewable energy (VRE) sources, or the grid operator). Box plot elements: Centre line: median, box limit: upper and lower quartiles, whiskers: 1.5x interquartile range, points: outliers **c**, Shares of solar PV depending on the learning rate of onshore and offshore wind energy, **d**, depending on the learning rate of solar PV and **e**, depending on the lead time for solar projects.

environments, such as macroeconomic conditions, business confidence, policy uncertainty and regulatory frameworks impact risk perceptions and the willingness to invest by domestic and international actors[13]. Equity investors and financial lenders apply high-risk premiums in perceived risky regional contexts, thus increasing the cost of capital for renewable projects[13,43].

Developing countries are particularly financially and fiscally constrained. Domestically they are characterised by under-developed capital markets and lack capital stock[44]; whereas international finance is restricted due to high sovereign risks and local currency risks on account of volatile economic fundamentals (as projects are funded with foreign currency while returns are generated in local currencies[45,46]). This leads to a chronic lack of available finance to support investments in solar energy.

Energy sector deficiencies further exacerbate the negative investment outlook for solar projects. Weak contract enforcement, changing energy regulations, and underdeveloped electricity markets affect project returns and investment viability. Developing countries may also face high import costs due to shortages in foreign currency reserves needed to support an expanding solar sector.

Consequently, a key challenge for global solar deployment lies in the mismatch between high investment needs (see Fig. 6 for modelled investment needs) and finance flows mobilised in developing countries[44]. Latest estimates suggest that climate financial flows would need to increase by a factor 4 to 8 in most vulnerable countries (IPCC 2022)[47]. Strategies to address this finance gap should include mechanisms to absorb currency and investment risk as a bridge to unlock international capital flows while creating domestic financial capacity over time.

As a third barrier, we discuss supply chains. A solar-dominated future is likely to be metal and mineral-intensive[48]. Future demand for "critical minerals" will increase on two fronts: electrification and batteries require large-scale raw materials – such as lithium and copper; niche materials, including tellurium, are instrumental for solar panels[49]. As countries accelerate their decarbonisation efforts, renewable technologies are projected to make up 40% of total mineral demand for copper and rare earth elements, between 60 and 70% for nickel and cobalt, and almost 90% for lithium by 2040[14].

The notion of criticality comes in three forms: physical, economic, and geopolitical. Firstly, there are risk associated with low reserves. Secondly, minerals supply typically reacts slowly to short-term changes in demand in, due to the long times required to establish mineral supply chains. This could lead to price rallies. The construction of new mining facilities (from exploration to mine operations) requires on average 16.5 years[14] and may be stalled due to concerns about socio-environmental impacts[50].

The geopolitical supply reliability of critical minerals is also weak, since mineral production displays higher geographical concentration, compared to fossil fuels production. China and The Democratic Republic of Congo, for example, own 60% and 70% of current global production of rare earth minerals and cobalt respectively[51]. Domestic shocks, including growing climate risks and political instability, could hamper the extraction and production and generate price shocks that along the value chain, impacting solar technology costs. Electricity networks could suffer similar impacts for nickel and aluminium.

Risk associated with low reserves can be mitigated with (research into) substitutions[52]. Recycling and circular economy processes can further reduce extraction rates, but re-used materials are unlikely to meet future demand as it outgrows existing stocks[53].

Lastly, resistance from declining industries may impact the transition. The pace of the transition depends not only on (economic) decisions by entrepreneurs, but also on how desirable policy makers consider it. Solar energy aligns with many policy objectives (clean air, poverty alleviation, energy security[54]). It also has disadvantages for some of the players involved, as it leads to rapid economic and industrial change.

Solar and wind power have a low energy density compared to alternatives. In most countries, they can provide enough energy to meet demand. However, land for renewables may be scarce close to population centres in some parts of the world[55,56]. Political tension on the use of land and water (for floating photovoltaics[57]) may increase as solar shares rise.

A rapid solar transition may also put at risk the livelihood of up to 13 million people worldwide working in fossil fuel industries and dependent industries. These people are frequently concentrated in communities close to mines extraction and industrial sites, where the

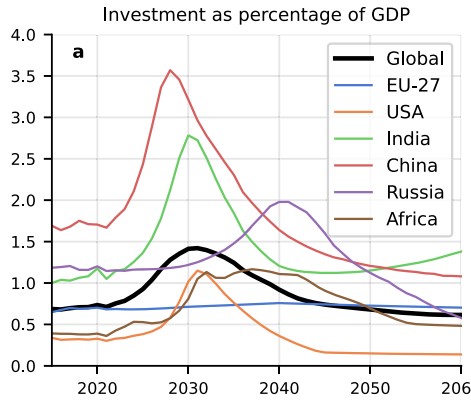
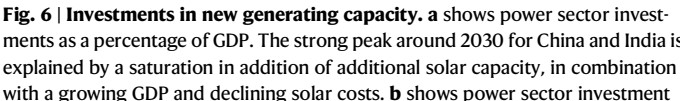
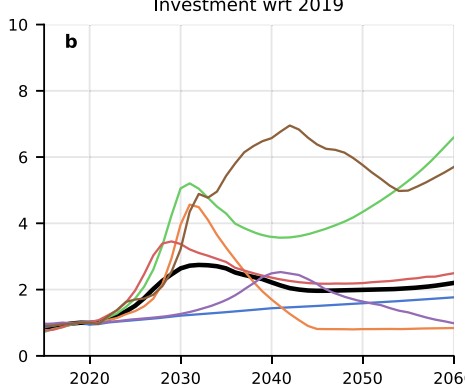

**Fig. 6 | Investments in new generating capacity. a** shows power sector investments as a percentage of GDP. The strong peak around 2030 for China and India is explained by a saturation in addition of additional solar capacity, in combination with a growing GDP and declining solar costs. **b** shows power sector investment with respect to 2019 values. Investment is forecast to see a fast growth worldwide relative to historical trends. Various regions in the Global South, in particular India and Africa, will see an even steeper rise in investment in generating capacity by mid-century, due to projected rapid economic growth.

closure of these activities can have severe repercussion on the well-being of communities decades on[58]. Policy makers could have substantial incentives to slow down the transition to limit these direct impacts. Similarly, many countries currently provide fossil fuel subsidies to increase the purchasing power of low-income households, difficult to phase out and which reinforce opposition to change. New coalitions of actors who benefit from the transition (home and land-owners, people with jobs in clean energy), may counterbalance some of the resistance from incumbents[15], but do not resolve equity issues. Regional economic and industrial development policy can resolve inequity, and can mitigate risks posed by resistance from declining industries[59].

## Discussion

Without any further energy policy changes, solar energy appears to follow a robust trajectory to become the future dominant power source before mid-century. Due to the reinforcing co-evolution of technology costs and deployment, our analysis establishes quantitative empirical evidence, from current and historical data trends, that a solar energy tipping point is likely to have passed. Once the combined cost of solar and storage crosses cost parity with all alternative technologies in several key markets, its widespread deployment and further costs declines globally could become irreversible. This echoes the results from Way et al.[16], who showed that such a configuration would be cheaper than alternatives[60].

A tipping point towards solar dominance however does not solve climate change mitigation or achieve climate targets, as it does not ensure a zero-carbon energy system. Solar-dominated electricity systems could become locked into configurations that are neither resilient nor sustainable with a reliance on fossil fuel for dispatchable power. Issues that could hinder achieving zero-carbon energy systems include grid stability issues, the availability of financial capital and critical minerals, and the willingness of decision-makers to get onboard a rapid transition that could generate substantial distributional issues in their respective regions. The energy crisis resulting from the war in Ukraine suggests that the accelerated move away from fossil fuels is needed even more urgently.

We conclude that achieving zero-carbon power systems likely requires policies of a different kind than have traditionally been discussed by the energy modelling community. The carbon price required to achieve cost break-even between renewables and fossil fuels may soon be zero. Instead, it is policies that address the above barriers—grid resilience, access to finance, management of material supply chains and political opposition—that may enable success in reaching net-zero energy emissions.

## Methods

E3ME-FTT-GENIE[61] is a model based on path-dependent simulation parameterised by historical data and technology diffusion trajectories. Integrated assessment models are typically based on utility or whole-system cost optimisation. Those models have played an important role in the energy debate by characterising what an optimal composition of the energy system ought to look like. They are less suitable for studying trends in energy system dynamics since, being driven by a centralised social planner construct, they neglect historical relationships, economic causality structures and decision-making processes[35,62]. In contrast, path-dependent energy system and economy simulations model system evolution on the basis of known causality structures and decision-making parameterised by timeseries and other data, however they do not identify optimal system configurations or policy. Decision-making by investors does not always line up with an optimal system, as investors use shorter time-scales to evaluate decisions compared to a putative 'social planner'.

In this paper, we use the energy-economy-environment (E3) simulation model E3ME-FTT-GENIE. It is grounded in empirically derived relationships between economic and technology variables, under the highest sectoral and regional disaggregation available for a global model (43 sectors and 70 regions) and a large number of energy technologies (88 technologies). Evolutionary dynamics form the core of technology evolution where induced innovation plays an important role; those sectors are represented by the various FTT submodels, which portray the typical S-shaped dynamics of technology uptake[63]. The model includes energy markets for nonrenewable and renewable energy. The GENIE climate and carbon cycle model is soft-coupled – emissions from E3ME-FTT drive the GENIE, but the GENIE does not affect the global economy. A complete set of equations for the E3ME-FTT model is given in Mercure et al.[61], with updates for the Power model found in Simsek et al.[41].

### FTT

The Future Technology Transformation (FTT) family of models provide an in-depth representation of four climate-relevant sectors in which technological change plays an important role: power, transport[64], heating[65] and steel[66]. These are the four energy end-use sectors with the highest greenhouse gas emissions. The models are based on evolutionary dynamics, simulating the S-curve of technology uptake characteristic of innovation[23]. Its core is the replicator dynamics equation (known as the Lotka-Volterra equation), prominent in ecosystem population dynamics modelling[67].

The direction of diffusion of a technology in FTT is primarily driven by comparing the levelised cost of technology options in chains

of binary discrete choice models, where the frequency of choice options availability is weighted by the share of those options in the technology mix. The levelised costs being compared are designed as to be a suitable depiction of decision making in each specific sector. A factor is included in each levelised cost, that captures non-pecuniary aspects otherwise not be captured with available data on costs alone. These are calibrated to match observed diffusion trajectories for each technology. For instance, technologies that are more socially attractive than their market costs suggest will have a negative factor included in the LCOE.

**FTT.** Power represents the diffusion of 24 technologies in the power sector. It includes nuclear, a set of bio-energy technologies, seven technologies based on the combustion of fossil fuels (including CCS options). Onshore and offshore wind, solar PV and CSP, hydro power, tidal, geothermal and wave power are also represented. FTT:Heat depicts the competition between various combustion technologies (oil, coal, wood and gas- burning) in households, as well as electrified heating options (resistive electric heating and heat pump technologies) and finally district and solar heating. FTT:Transport models the competition between petrol, diesel, LPG, EVs and hybrid passenger vehicles, as well as motor vehicles. For each base technology, there is a further disaggregation based on the luxury of the vehicle. Finally, FTT:Steel models 25 different routes of steel production: on the basis of coal, gas, hydrogen and electricity.

### FTT:Power

FTT Power follows Ueckerdt et al.[68] in its detailed representation of variable renewables in grid stability. Technologies are classified along six load bands, and production is allocated to available technologies based on intermittency and flexibility constraints. This takes into account the hourly demand over time in a set of key regions, and hourly supply potential per technology. For each mix of variable renewables, the optimal curtailment and storage needs are estimated using the parametrizations from Ueckerdt et al.[68]. Compared to earlier treatment in FTT, this implies much improved and less conservative assumptions over limits to renewables in power grids due to intermittency[41].

The baseline scenario (the only scenario in this paper) includes the EU Emission Trading System explicitly, as well as the ongoing nuclear phase-out in Germany and Belgium. Other policies are included implicitly by adding "gamma values" to the LCOE values used for decision-making by investors. These gamma values are calibrated to produce short-term projection of power capacity shares in each country that is consistent with the recent historical trend, by minimising the difference in rate of growth or decline at the changeover point between history and simulation. As a conservative assumption, we do not include a premature retirement of power plants when their marginal costs rise above the LCOE of newly installed power plants. We also do not include the possibility to extend the lifetimes of power plants.

The CAPEX and OPEX costs are derived from the IEA's *Projected Costs of Generating Electricity 2020*, and medians are used to fill in missing data. For solar, we use utility-scale solar prices. Residential solar power is more expensive, but the attractiveness for consumers is heightened by the fact they avoid various taxes on electricity. Standard deviations of these costs are also derived from this dataset; this means that volatility over time is not captured in our uncertainty.

This paper includes a further set of updates to FTT:Power that collectively favour the diffusion of solar PV into the electricity mix. Based on historical data from BNEF (see Supplementary Figure 4), we introduce learning in operational costs, rather than only in CAPEX, which mostly benefits offshore wind and solar PV. Learning rates are updated for key technologies, following Way et al.[8]. Both solar power

and wind energy see a higher learning rate than previous model versions. Based on recent estimates of panel lifetime, we assume that a solar panel lasts 30 years on average.

Using BNEF data up to 2020, through a whole-model data upgrade, we update realised capacity factors for onshore, offshore, and solar technologies to the most recent values. The timescale for developing offshore wind projects is found to be longer than onshore wind, which hinders rapid growth.

The technical potential for onshore wind is updated using[69], which has an improved resolution, threshold wind speed and turbine technical specifications compared to[70]. For solar power (solar PV and CSP), we updated the technical potential as the sum of[71] (utility-scale solar) and[72] (rooftop solar). We did not include a technical potential[57] for application of solar power on water ("floatovoltaics"), as this technology is still in early stages of development.

Regions with offshore potential, but no installed capacity, are attributed a small offshore wind capacity, equal to 1/100 the capacity of onshore wind installed in the region or country. Similar seeding is performed for CSP, which equals 1/100 the capacity of solar in the country. For countries without any onshore capacity, a small capacity, equal to 0.1% of historical generation, is added. This is because technology with zero deployment will never be selected. Historical installed capacity of renewables is inserted using[9].

We innovate by introducing learning in storage technologies, which were, in the original model, fixed at the estimated 2030 price levels. For short-term storage we take the average of the learning rate for lithium-ion batteries and vanadium flow batteries. The latter are less common currently, but provide more flexibility and have a lower environmental impact[73]. The averaged learning exponent is 0.255 and long-term storage (assumed to be supplied by hydrogen) a more modest learning of 0.194 based on[8]. System storage costs are divided over the variable renewables. Both short-term storage costs and long-term storage costs increase with a poorer ratio between sun and wind. CSP only contributes to long-term storage costs, as it contains short-term storage internally. This is a conservative assumption for variable renewable energy diffusion, as policy may attribute storage costs to all grid participants or directly to customers.

The uncertainty analysis of Fig. 3, Fig. 5 and Supplementary Figure 5 is performed with a Monte Carlo sampling of a set of input parameters. Input parameters were selected that had the largest expected impact on the diffusion of power generation technologies. In half of cases, costs of power storage were attributed equally among participants in the power market, whereas the costs of storage were allocated to renewables in the other half (the default). Inequality around access to capital between countries was modelled via the discount rate: the costs of finance (WACC/discount rate) was varied between 0.075 and 0.100 for countries in the OECD, and varied between 0.100 and 0.125 for all other countries. The learning rates for solar and wind were varied per the distribution given in Way et al.[8]. The importance of nonpecuniary aspects (gamma values), captured using calibration, was multiplied by a value drawn from the normal distribution $N(1, 0.2)$. Similarly, fuel costs for gas and coal were varied by a factor drawn from the same normal distribution. Possible delays in grid expansion (f.i. to resolve grid congestion) are expressed as increasing the lead time of solar PV development with a Poisson distribution. The lifetime of solar panels was varied uniformly between 25 and 35 years.

### E3ME

The E3ME model is the macro-econometric component of the modelling framework. It is demand-led and features 70 regions and countries, covering the world. Each EU member and the UK has a representation of 70 sectors; other regions are represented to 43 economic sectors. The sectors are linked with input-output tables, and

bilateral trade equations link the various regions and countries. The energy system within the E3ME model consists of equations for 23 fuel users (for instance chemical industry or air transport), and 12 fuel types (for instance electricity, or crude oil). Fifteen econometric regressions calibrated on data from 1970 to 2019 form the basis of the model. The model can be extended up to 2070. As a demand-led model, it first computes demand for final goods and services, and the supply of intermediate goods is estimated using input-output tables and bilateral trade relationships, which then drive employment, investment, income, induced productivity change, price levels and other macro variables[63].

The IO tables are dynamically coupled to the FTT models via the energy balances. In specific, the coefficients for coal, oil and gas, manufactured fuels, electricity and "gas, steam & air conditioning" are adjusted based on the outcome of the FTT models.

The model uses World Bank estimates of historical GDP growth, the UN World Population Prospects for demographic change, and the IEA Energy Balances for energy demand growth, and the World Energy Outlook for baseline future energy demand[74]. The model does not incorporate the SSP framework, but our baseline can be compared most closely to SSP2 (the "middle of the road scenario"[38]. Philosophically, the model and SSP2 have the same narrative: a continuation of current trends.

Population growth in our model is slightly higher than SSP2, but economic growth is lower compared to SSP2 in many major economies (but always above SSP3). Total primary energy demand in 2050 is very similar to SSP2. For more details, see[61].

## Data availability
Historical generation and capacity of renewable energy from IRENA is available at https://irena.org/publications/2022/Apr/Renewable-Capacity-Statistics-2022. Original data from BNEF and IEA are licensed by these owners, but datasets derived by the authors are available as part of the model code (see code availability). Source data for the figures are provided with this paper at https://doi.org/10.6084/m9.figshare.22659052.

## Code availability
The code for the standalone FTT model can be found at https://github.com/cpmodel/FTT_StandAlone/tree/Is_a_solar_future_inevitable[75]. This version was used for the uncertainty quantification of Figs. 3 and 5. The computer code for the full E3ME-FTT model needed to replicate the study is licensed and not available publicly, but can be obtained from the authors upon reasonable request.

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

## Acknowledgements

JFM, FJMMN and PV received funding from the UK Department for Business, Energy and Industrial Strategy (BEIS) from the EEIST project. NA acknowledges support from the European Research Council (ERC) under the European Union's Horizon 2020 research and innovation programme (grant agreement No 802891), a grant which also funded SK., FL and JR. We would like to thank Simon Sharpe for discussion that improved the discussion on policy implications. We would also like to thank Doyne Farmer for his valuable feedback.

## Author contributions

F.J.M.M.N. coordinated and performed the research, with contributions from J.-F.M., and N.A. F.J.M.M.N and J.-F.M. wrote the article with support from N.A., S.K. and F.L. N.A., F.L, S.K., and J.R. collected the BNEF data. F.J.M.M.N. led the model improvements and ran the simulations, with support from P.V., J.-F.M and H.P.

## Competing interests

The authors declare no competing interests.
