## [Peer Review File · Nature Communications]

Is a solar future inevitable?Reviewers' Comments:

Reviewer #1:

Remarks to the Author:

Overall, I think this article is clear, well written, well argued, topical, of consistently high quality, and certainly worthy of publication. My comments are intended mainly as suggestions and thoughts for the authors to consider (but not necessarily feel obliged to act on) in the final version.

General Comments

The evidence, approach, and resulting insights in the paper are very similar to Way et al. in Joule (ref 10 but needs updating to published version) albeit with a different systems modelling tool. The main difference in this paper seems to be the discussion of barriers to large-scale RE diffusion, and the sensitisation of results to those barriers (which Way et al. also reflect on to some extent in their discussion) It would be helpful if the authors could explain clearly how and why their work builds on, differs from, or challenges the results of Way et al., and why the different systems modelling approach yields similar (or different) insights.

Specific Comments

The Word doc manuscript I read had no page or line numbers. I am using the first page with title, abstract, and the beginning of the introduction as p1.

p2 "the rapid disruptive transition" - applies specifically to electricity systems rather than the wider energy system (although clearly these are interlinked). The article generally does a good job of referring specifically to electricity rather than generically to energy, but it would be good if the authors could do this consistently throughout.

p2 "a series of barriers" - misses also the large implied land (or rooftop) requirements for massive solar scale up given its very low energy densities (and ditto wind).

p2 "underestimated uptake rates of solar energy" - one of many good examples of a topic covered exhaustively in Way et al. (not referenced here)

p3 "solar ... high learning rates ... modularity ... replicability" - both Sweerts et al. 2020 and Wilson et al. 2020 show this for diverse sets of energy technologies as a function of unit size and modularity, and these were nicely summarised in IPCC AR6 too

p3 "historical failure ... " - exogenously imposed diffusion constraints specifically designed to restrain overly rapid technological shifts (to RE) also played a part, particularly in optimisation frameworks

p3 "we use IEA, BNEF ... " - to calibrate what?

p3 "long lifetimes prevent ..." - without large-scale premature retirement of capital stock (prior to the end of its technical lifetime) or stranded assets, a topic which would be welcome to revisit in the discussion

Fig 1 - this is a niggle, but onshore and offshore would benefit from also being wind!

Fig 1 and explanatory text below - there's no information given on the many other assumptions behind these projections; assumptions which the SSP framework systematically and transparently varies given baseline uncertainties (e.g., rates of economic growth, demographic change, energy service demand growth, electrification rates to account for electrification of heat and transport, and so on). Can you clarify in brief here, and in detail in Methods, the broader assumptions used in the

projections. This absolutely does not have to be linked to the SSP framework, but how the projections position approximately within the SSP framework would then be helpful to compare and contrast results with the large SSP-informed literature.

p5 "learning rate for long-duration storage" - I wasn't always clear what was being referred to here: is it green H2 for bridging inter-seasonal variation? inter-seasonal or even inter-annual storage is an important issue for high RE grids, so this would merit being spelt out more clearly

Fig 3 - this is a really nicely designed Figure, illustrating a key argument with the viewer in mind ... but would gas not be a better comparator than coal, as gas is dominant, growing, and sets the marginal price in many markets while coal is already in decline in many jurisdictions (+ onshore WIND!)

p6 "72% of simulations" - clarify these are the full set of simulations used for robustness testing

p6 "rules out a subset of SSPs" - these need explaining (for the uninitiated reader) + see my earlier points on it being hard to interpret this comment as no equivalent information to SSP storylines is given for your projections (other than on technology costs)

p7 "suggests that further climate policy ... " - I think this is a key and important insight from the analysis, and is rightly emphasised again in the discussion ... but it's corollary is that any and all remaining price support for RE can be stopped without causing a meaningful dent in uptake rates: is this borne out in jurisdictions that have done this? and does it still hold if cost-related climate policy support for RE is shifted to other large-scale low-carbon options like CCS or nuclear?

p8 "notably, new power market rules can be designed ..." - I didn't understand this point, can you exemplify or explain what generators are that diversify intermittency

p8 "charged directly to consumers ..." - I think more generally there are approaches that require individual generators to pay for intermittency risk one way or the other, versus approaches that collectivise or socialise intermittency risk (through bills, utility charges, regulated requirements, etc.)

Fig 6 - why is there a strong 2025-2030 peak, what explains this?

Fig 6 legend - "moderate growth worldwide" looks like a tripling in 10 years ... is this moderate??

p10 "utility-scale requires land ... may be scarce near population centres" - see earlier point, this is quite a throwaway mention of a potentially important constraint. Can you justify your confidence this won't be a constraint (e.g., in reference to spatially-explicit RE diffusion studies with equivalent TW deployments to your projections?)

p10 conclusion - first para echoes Way et al., see general comment about explaining value added of your study

p10 "issues that could hinder ..." - as well as land, is there not relevant insight into northern versus southern latitudes and output per installed capacity under different insolation regimes? is this partly why Russia seems the laggard region (relatively speaking)

p11 "GENIE ... soft-coupled; affected by the global economy ... " - weird grammar so a bit hard to follow: basically emissions from FTT-E3ME drive climate model, but climate impacts do not feedback into FTT-E3ME + ref for Mercure et al. missing

p12 "similar seeding is performed for CSP ... " - is this because a technology with zero deployment will never be selected? please clarify

p12 "long-term storage a more modest learning ... " - see earlier point, please be clearer about what this long-term storage is

p13 E3ME - how are static IO tables in E3ME consistent with dynamic representations of technological substitution in FTT? (or are the IO tables dynamically coupled into the FTT model in some way)

p14 GENIE - I couldn't see how GENIE was used in this study as there's no climate results reported ... it would actually be interesting to see what the warming outcome is of the reference projections, relative to the scenario literature

References - need a good cleaning and checking: e.g., refs 18 and 21 are the same; are refs 12 and 34 also the same?

References

Sweerts, B., R. J. Detz and B. van der Zwaan (2020). "Evaluating the Role of Unit Size in Learning-by-Doing of Energy Technologies." *Joule* 4(5): 967-970. DOI: 10.1016/j.joule.2020.03.010

Way, R., M. C. Ives, P. Mealy and J. D. Farmer (2022). "Empirically grounded technology forecasts and the energy transition." *Joule* 6(9): 2057-2082. DOI: <https://doi.org/10.1016/j.joule.2022.08.009>

Wilson, C., A. Grubler, N. Bento, S. Healey, S. De Stercke and C. Zimm (2020). "Granular technologies to accelerate decarbonization." *Science* 368(6486): 36-39. DOI: 10.1126/science.aaz8060

Reviewer #2:

Remarks to the Author:

Thank you for giving me the opportunity to read this paper. I believe the paper makes an important contribution to the field of renewable energy adoption, using the lens of learning and diffusion models. I have several suggestions for improvements:

The authors should explain how they have identified and selected the four barriers or bottlenecks to diffusion of renewable/solar energy, including grid resilience, investment barriers, supply chains, and political economy, and why not others.

The authors could clarify whether their calculations are based on residential solar PV or utility-scale solar PV. The LCOE will be different for these two types of solar PV.

A discussion could be added on the role of concentrated solar power. Studies show that solar PV is deployed earlier due to sharp price decreases in the last decades and that concentrated solar thermal technology will gain a competitive advantage when PV has a high market share and its system integration costs become too high (e.g., Pietzcker et al., 2014). What role could concentrated solar power play in grid resilience?

The authors state that: "no mechanism guarantees that optimal grids are achieved if left to market forces, especially in contexts of diverging technology costs, and solar dominance could become self-limiting." The authors could consider the fact that renewable energy companies will have an incentive to invest in the grid (and they do invest in grid and storage technology) to enable their industry to overcome barriers to solar and wind adoption.

The authors also state that: "our model suggests that the allocation of storage costs to the grid and charged directly to consumers incentivises more renewables diffusion than requiring each project to

provide their own storage". Regarding this statement I wonder what this suggestion does to the learning curves considered in the models and the LCOE that include system storage costs. Private RE actors who compete for solar projects innovate and invest in grid and storage technology and through competition drive down costs. Is the learning curve and the cost reduction faster for private actors than when we leave the investments up to (often) publicly owned national or regional monopolies for grid operation?

In barrier 2, does the suggestion on lines 241-243 refer to involvement of the World bank and other investment funds for developing countries, i.e., those institutions that European-based solar project developers already collaborate with for their solar projects in developing economies?

The conclusion could highlight the novelty of the results of this paper that result from using the simulation methods.

Minor comments:

The sentence on lines 96-98 requires a source, or an explanation as to why these are main reasons for the failure of existing models.

The sentence in line 130 requires more explanation; it is not clear how this follows from the above evidence or arguments.

The sentence in lines 133-135 seems to contradict itself; if the slow uptake is attributed to non-pecuniary aspects how would lower prices help?

Figure 1 could refer to offshore wind and onshore wind. Figure 4b needs to define VRE and a legend is needed to explain the numbers on the x-axis of figure 4e.

What is the source of figure 6? Number 42?

Pietzcker, R., Stetter, D., Manger, S., and Luderer, G. (2014). Using the sun to decarbonize the power sector: The economic potential of photovoltaics and concentrating solar power. *Applied Energy*, 135, 704-720.

REVIEWER COMMENTS

Reviewer #1 (Remarks to the Author):

Overall, I think this article is clear, well written, well argued, topical, of consistently high quality, and certainly worthy of publication. My comments are intended mainly as suggestions and thoughts for the authors to consider (but not necessarily feel obliged to act on) in the final version.

Thank you very much for the comments. We agree on the majority of points and have incorporated them into the paper. On the remaining points we have provided justification below'. The paper is now much clearer.

General Comments

The evidence, approach, and resulting insights in the paper are very similar to Way et al. in Joule (ref 10 but needs updating to published version) albeit with a different systems modelling tool. The main difference in this paper seems to be the discussion of barriers to large-scale RE diffusion, and the sensitisation of results to those barriers (which Way et al. also reflect on to some extent in their discussion) It would be helpful if the authors could explain clearly how and why their work builds on, differs from, or challenges the results of Way et al., and why the different systems modelling approach yields similar (or different) insights.

There are two key differences Way et al and our work. The first is that our model contains the feedback between investment and learning endogenously. The price declines impact our investment decision making core, which further drive price declines. Way et al take a scenario perspective, where investments are chosen exogenously.

The second one is that our model is disaggregated into 70 world regions. As such, we can track how uptake of renewables is likely to evolve in countries with a wide variety of circumstances in terms of technical potential, variability of production for VRE and pricing.

We've added a paragraph detailing how the studies use complementary methodologies to reach a similar conclusion.

Specific Comments

The Word doc manuscript I read had no page or line numbers. I am using the first page with title, abstract, and the beginning of the introduction as p1.

p2 "the rapid disruptive transition" - applies specifically to electricity systems rather than the wider energy system (although clearly these are interlinked). The article generally does a good job of referring specifically to electricity rather than generically to energy, but it would be good if the authors could do this consistently throughout.

Done

p2 "a series of barriers" - misses also the large implied land (or rooftop) requirements for massive solar scale up given its very low energy densities (and ditto wind).

Thank you for the comment. We provide justification below (reviewer 2 report).

p2 “underestimated uptake rates of solar energy” - one of many good examples of a topic covered exhaustively in Way et al. (not referenced here)

Cited in addition to Meng et al

p3 “solar ... high learning rates ... modularity ... replicability” - both Sweerts et al. 2020 and Wilson et al. 2020 show this for diverse sets of energy technologies as a function of unit size and modularity, and these were nicely summarised in IPCC AR6 too

Now cited both Sweerts and Wilson, and removed one of the less relevant citations.

p3 “historical failure ... “ - exogenously imposed diffusion constraints specifically designed to restrain overly rapid technological shifts (to RE) also played a part, particularly in optimisation frameworks

Added and cited to Way et al.

p3 “we use IEA, BNEF ... “ - to calibrate what?

Clarified: “We use IEA data for historical generation, CAPEX and OPEX, BNEF for capacity factors, construction and lifetimes until 2020, IRENA for historical renewables capacity data between 2019 and 2021 and 2021”

p3 “long lifetimes prevent ...” - without large-scale premature retirement of capital stock (prior to the end of its technical lifetime) or stranded assets, a topic which would be welcome to revisit in the discussion

Our paper sketches a baseline, a likely scenario when policies are unchanged. It may distract to mention premature retirement of capital stock here too, which usually occurs in scenarios with more explicit phase-outs of fossil-fuels. A more nuanced description of stranded assets can be found in other work with this model (f.i., Semieniuk et al. 2022).

Fig 1 - this is a niggle, but onshore and offshore would benefit from also being wind!

Added

Fig 1 and explanatory text below - there’s no information given on the many other assumptions behind these projections; assumptions which the SSP framework systematically and transparently varies given baseline uncertainties (e.g., rates of economic growth, demographic change, energy service demand growth, electrification rates to account for electrification of heat and transport, and so on). Can you clarify in brief here, and in detail in Methods, the broader assumptions used in the projections. This absolutely does not have to be linked to the SSP framework, but how the projections position approximately within the SSP framework would then be helpful to compare and contrast results with the large SSP-informed literature.

We included a rough comparison to the SSP framework in the methods section, including information about economic growth and demographic change, and added two references that describe our assumptions in more details.

p5 “learning rate for long-duration storage” - I wasn’t always clear what was being referred to here: is it green H2 for bridging inter-seasonal variation? inter-seasonal or even inter-annual storage is an important issue for high RE grids, so this would merit being spelt out more clearly

Done. We assume hydrogen is used for long-term storage.

Fig 3 - this is a really nicely designed Figure, illustrating a key argument with the viewer in mind ... but would gas not be a better comparator than coal, as gas is dominant, growing, and sets the marginal price in many markets while coal is already in decline in many jurisdictions (+ onshore WIND!)

Thank you for the compliment. Over the last few years, the price of gas has made many large jumps and falls, and it is a bit unclear how this is changing in the future. We fear that adding gas to the figure will lead to some arbitrariness. Changed it to ONSHORE wind.

p6 “72% of simulations” - clarify these are the full set of simulations used for robustness testing

Done. These are the full set.

p6 “rules out a subset of SSPs” - these need explaining (for the uninitiated reader) + see my earlier points on it being hard to interpret this comment as no equivalent information to SSP storylines is given for your projections (other than on technology costs)

We have included a discussion on how the model compares with SSP2 in the methods section, and explained what SSPs are in the main text, and what type of SSPs we rule out (the ones with increasing coal shares, such as SSP5).

p7 “suggests that further climate policy ... “ - I think this is a key and important insight from the analysis, and is rightly emphasised again in the discussion ... but it’s corollary is that any and all remaining price support for RE can be stopped without causing a meaningful dent in uptake rates: is this borne out in jurisdictions that have done this? and does it still hold if cost-related climate policy support for RE is shifted to other large-scale low-carbon options like CCS or nuclear?

We have not researched if remaining price support for RE can be stopped, but there are examples of countries phasing out monetary support for renewables without it affecting uptake (for instance, offshore wind in the Netherlands and Germany (Rentier et al., 2023)).

Countries with small solar and wind industries would likely benefit from keeping subsidies to speed up deployment. Furthermore, alternative RE technologies to wind and solar PV (such as CSP, geothermal and small hydro) are often not attractive to investors, and may play an important role in grid stabilisation, so subsidies remain important for those technologies. The difference in LCOE between nuclear and coal+CCS and solar is quite big. It may be more likely that these will replace fossil fuels plants without CCS if their support is increased, given a similar role they play in the grid.

p8 “notably, new power market rules can be designed ...” - I didn’t understand this point, can you exemplify or explain what generators are that diversify intermittency

Changed “that diversify grid sources of intermittency” with “that complement solar production on a daily to seasonal scale”

p8 “charged directly to consumers ...” - I think more generally there are approaches that require individual generators to pay for intermittency risk one way or the other, versus approaches that collectivise or socialise intermittency risk (through bills, utility charges, regulated requirements, etc.)

Agreed. We investigate a variety of payment options in a separate paper (in preparation) and consider it further discussion out of scope for this paper.

Fig 6 - why is there a strong 2025-2030 peak, what explains this?

The 2025-2035 peak coincides with the fastest-growing part of the S-curve. This is a combination of two factors: firstly, we have only recently reached the point that renewables are cheaper than alternatives, so that investing in renewables makes economic sense. But, the industry in many countries is still growing, and funds there are limits to how fast knowledge and technology can build out, so the peak only happens a few years into the future.

Fig 6 legend - “moderate growth worldwide” looks like a tripling in 10 years ... is this moderate??

No, not really, good point. Changed to fast.

p10 “utility-scale requires land ... may be scarce near population centres” - see earlier point, this is quite a throwaway mention of a potentially important constraint. Can you justify your confidence this won't be a constraint (e.g., in reference to spatially-explicit RE diffusion studies with equivalent TW deployments to your projections?)

We agree we wrote that too confidently, and have now dedicated a paragraph to the issue. We believe it is less of a physical barrier, but rather a political barrier, as the technical potential is available in a large majority of regions. Our model incorporates a maximum technical potential, using the sum of residential technical potential and utility-scale technical potential. These numbers are derived from sources with an explicit spatial component. We are disregarding the technical potential of “floatovoltaics” (<https://www.nature.com/articles/d41586-022-01525-1>), which can fully provide solar demand in some countries, but is a newer application.

p10 conclusion - first para echoes Way et al., see general comment about explaining value added of your study

We've included more references to Way et al, and add a paragraph that explains how, with quite different methodology, we reach similar conclusion.

p10 “issues that could hinder ...” - as well as land, is there not relevant insight into northern versus southern latitudes and output per installed capacity under different insolation regimes? is this partly why Russia seems the laggard region (relatively speaking)

On a global scale, we do not see this as a major barrier. It is true of course that solar is less attractive in Northern latitudes, and our model has limited uptake of solar in Scandinavian and Baltic countries. But already in countries as far north as Germany, we model a significant role for solar. Southern Russia has a good solar potential, but still very little solar production. The abundance of cheap nuclear and fossil fuels is a larger barrier there than insolation per se.

1 “GENIE ... soft-coupled; affected by the global economy ... ” - weird grammar so a bit hard to follow: basically emissions from FTT-E3ME drive climate model, but climate impacts do not feedback into FTT-E3ME + ref for Mercure et al. missing

Reworded and citation added.

p12 "similar seeding is performed for CSP ... " - is this because a technology with zero deployment will never be selected? please clarify

Indeed, clarified

p12 "long-term storage a more modest learning ... " - see earlier point, please be clearer about what this long-term storage is

Clarified. It is assumed to be hydrogen.

p13 E3ME - how are static IO tables in E3ME consistent with dynamic representations of technological substitution in FTT? (or are the IO tables dynamically coupled into the FTT model in some way)

The IO tables are dynamically coupled to the FTT models via the energy balances. In specific, the coefficients for coal, oil and gas, manufactured fuels, electricity and "gas, steam & air conditioning" are adjusted based on the coupling with the FTT models. I've added this to the methods section.

p14 GENIE - I couldn't see how GENIE was used in this study as there's no climate results reported ... it would actually be interesting to see what the warming outcome is of the reference projections, relative to the scenario literature

I have removed the paragraph on GENIE. We do indeed not report on outcomes from GENIE in this paper. I have chosen not to include a temperature projection, as some key parts of the model (heat/transport) are undergoing major improvements, so this number would possibly not be very realistic.

References - need a good cleaning and checking: e.g., refs 18 and 21 are the same; are refs 12 and 34 also the same?

18 and 21 now separated. Refs 12 and 34 were by the same author, but not the same.

References review

Sweerts, B., R. J. Detz and B. van der Zwaan (2020). "Evaluating the Role of Unit Size in Learning-by-Doing of Energy Technologies." *Joule* 4(5): 967-970. DOI: [10.1016/j.joule.2020.03.010](https://doi.org/10.1016/j.joule.2020.03.010)

Way, R., M. C. Ives, P. Mealy and J. D. Farmer (2022). "Empirically grounded technology forecasts and the energy transition." *Joule* 6(9): 2057-2082. DOI: <https://doi.org/10.1016/j.joule.2022.08.009>

Wilson, C., A. Grubler, N. Bento, S. Healey, S. De Stercke and C. Zimm (2020). "Granular technologies to accelerate decarbonization." *Science* 368(6486): 36-39. DOI: [10.1126/science.aaz8060](https://doi.org/10.1126/science.aaz8060)

References in answer:

Rentier G, Lelieveldt H, Kramer G (2023) Institutional constellations and policy instruments for offshore wind power around the North sea. *Energy Policy* 173

Semieniuk, Gregor, Philip B. Holden, Jean-Francois Mercure, Pablo Salas, Hector Pollitt, Katharine Jobson, Pim Vercoulen, Unnada Chewpreecha, Neil R. Edwards, and Jorge E. Viñuales. 2022. "Stranded Fossil-Fuel Assets Translate to Major Losses for Investors in Advanced Economies." *Nature Climate Change* 12 (6): 532–38. <https://doi.org/10.1038/s41558-022-01356-y>.

Reviewer #2 (Remarks to the Author):

Thank you for giving me the opportunity to read this paper. I believe the paper makes an important contribution to the field of renewable energy adoption, using the lens of learning and diffusion models. I have several suggestions for improvements:

Thank you for the detailed review comments. We agree with most comments and have incorporated them into the paper. On the remaining points we have provided an explanation of our reasoning below. The paper is now much clearer.

The authors should explain how they have identified and selected the four barriers or bottlenecks to diffusion of renewable/solar energy, including grid resilience, investment barriers, supply chains, and political economy, and why not others.

Thank you for the comment. Different barriers, bottlenecks and problems affect different renewable energy depending on the source, geographical contexts, country-specific market structures and levels of maturity (IRENA, 2019). We reviewed the literature and selected the four worldwide, high level barriers that builds upon the four levels (technological, economic, policy and socio-political) flagged in IRENA, 2019

Scholars have studied context-specific barriers that hamper successful scale-up of solar PV (Table below) focusing on non-technical (*soft*) barriers such as perceptions, information, access to capital and uncertainty (Meijer et al. 2019; Mah et al. 2018; Xue, Lindkvist, and Temeljotov-Salaj 2021; Rai, Reeves, and Margolis 2016; Do et al. 2020)) and on technical factors, including grid resilience, financing mobilisation, trade and geophysical constraints (IRENA 2014; Tong et al. 2021; Wang et al. 2021). Our four hampering factors (grid resilience, availability of finance, supply chain and the pace of the transition) interest and span across all identified and literature-informed barriers. Moreover, they build upon and expand the four core limitations described in IRENA (2019): technological, policy, market and economic and regulatory, political and social barriers.

Grid resilience encompasses technical, but also political and regulatory issues as policy mixes can overcome intermittency. Finance mobilisation interests economic, social and political barriers as a more equal distribution of financial resources will expand solar PV in emerging economies too. Supply chain barriers lead to geopolitical, policy and technological tensions and frictions as the critical mineral industry and technology providers in the renewable energy space will have to align and coordinate to satisfy future demand. Finally, the pace of the transition involves all four IRENA (2019) building blocks.

The authors could clarify whether their calculations are based on residential solar PV or utility-scale solar PV. The LCOE will be different for these two types of solar PV.

Clarified in methods (we use utility-scale pricing). In future work we plan to incorporate the higher residential LCOE and avoided taxes for consumers from producing their own electricity.

A discussion could be added on the role of concentrated solar power. Studies show that solar PV is deployed earlier due to sharp price decreases in the last decades and that concentrated solar

thermal technology will gain a competitive advantage when PV has a high market share and its system integration costs become too high (e.g., Pietzcker et al., 2014). What role could concentrated solar power play in grid resilience?

In our model, CSP plays a growing, but moderate role. We include the advantage of CSP as a dispatchable source, and only require it to pay for seasonal storage (as thermal storage with most CSP system precludes the need for short-term Lithium-ion storage). However, as we are investigating a scenario without additional policy, we may not fully capture the advantages of CSP: in a scenario with a fossil fuel phase-out, the role of CSP could become much larger. Small technologies usually do not replace established technologies without a targeted push, unless the advantages far outweigh the system's inertia. Since Pietzcker et al, the relative advantage of CSP versus solar PV has slightly diminished, as lithium-ion batteries have seen a tremendous drop in prices. I've added a short description to the results:

Concentrated solar power grows over the entire period, but without targeted policy its overall share in the power mix remains small, despite its advantage as a dispatchable source of electricity.

Our methodology relies strongly on historical trends, and in the last few years CSP has had little growth, and even a decline in 2021 (REN21, 2022).

The authors state that: "no mechanism guarantees that optimal grids are achieved if left to market forces, especially in contexts of diverging technology costs, and solar dominance could become self-limiting." The authors could consider the fact that renewable energy companies will have an incentive to invest in the grid (and they do invest in grid and storage technology) to enable their industry to overcome barriers to solar and wind adoption.

This is certainly true, but the extent to which they can make these investments depends on the market structure, which does not automatically align with a theoretical optimality. Market decisions are often made on short time scales. For instance, a grid dominated by solar would benefit from the presence of pumped hydro. However, the economics of pumped hydro are difficult in the current situation, and, in combination with market uncertainty, the long construction times would not allow for these plants to be built at an optimal time (MIT, 2022).

The authors also state that: "our model suggests that the allocation of storage costs to the grid and charged directly to consumers incentivises more renewables diffusion than requiring each project to provide their own storage". Regarding this statement I wonder what this suggestion does to the learning curves considered in the models and the LCOE that include system storage costs. Private RE actors who compete for solar projects innovate and invest in grid and storage technology and through competition drive down costs. Is the learning curve and the cost reduction faster for private actors than when we leave the investments up to (often) publicly owned national or regional monopolies for grid operation?

That's a very interesting question. In our model, learning curves are dependent on the cumulative deployment of a technology, independent on who deploys. I'm unaware of research that directly answer that question.

Winfield et al. discussed this question from a multi-level perspective approach¹. They concluded that liberalised and monopoly markets have different opportunities to go from niche-level innovation to a regime transition, and thereby to bring down costs. Liberalised markets will have niches created by a diversity of third-party investors. Monopolies can choose to create shielded niches for innovation to take place. The paper did not conclude that either option is better.

In barrier 2, does the suggestion on lines 241-243 refer to involvement of the World bank and other investment funds for developing countries, i.e., those institutions that European-based solar project developers already collaborate with for their solar projects in developing economies?

The World Bank has a relatively small balance sheet relative to the size of the problem, so while they play an important role, the overall scale of the challenge is much larger and should include the participation of a wide scale of financial institutions.

The conclusion could highlight the novelty of the results of this paper that result from using the simulation methods.

We have accentuated the difference from Way et al (who use statistical methodology) in the main text, and clarified the how the simulation method with “decision-making by investors” differs from a “social planner” optimisation method in the methodology section.

Minor comments:

The sentence on lines 96-98 requires a source, or an explanation as to why these are main reasons for the failure of existing models.

Provided citations to Way et al and Jaxa-Rosen et al, and amended text to remove “optimisation models” and add “imposition of maximum deployment levels” per these sources.

The sentence in line 130 requires more explanation; it is not clear how this follows from the above evidence or arguments.

Moved the sentence to after the uncertainty analysis, which support the sentence better.

The sentence in lines 133-135 seems to contradict itself; if the slow uptake is attributed to non-pecuniary aspects how would lower prices help?

Clarified: when prices fall sufficiently far under alternatives, the pecuniary aspects overcome the nonpecuniary aspects.

Figure 1 could refer to offshore wind and onshore wind. Figure 4b needs to define VRE and a legend is needed to explain the numbers on the x-axis of figure 4e.

What is the source of figure 6? Number 42?

Done. Figure 6 is based on model output, now clarified.

¹ Winfield, Shokrzadeh, and Jones, ‘Energy Policy Regime Change and Advanced Energy Storage’.

Citations review

Pietzcker, R., Stetter, D., Manger, S., and Luderer, G. (2014). Using the sun to decarbonize the power sector: The economic potential of photovoltaics and concentrating solar power. *Applied Energy*, 135, 704-720.

Citations answers:

REN21. 2022. Renewables 2022 Global Status Report. Paris: REN21 Secretariat. ISBN 978-3-948393-04-5

IRENA (2019), *Global energy transformation: A roadmap to 2050* (2019 edition)

Do, Thang Nam, Paul J. Burke, Kenneth G. H. Baldwin, and Chinh The Nguyen. 2020. "Underlying Drivers and Barriers for Solar Photovoltaics Diffusion: The Case of Vietnam." *Energy Policy* 144 (September): 111561. <https://doi.org/10.1016/j.enpol.2020.111561>.

IRENA. 2014. "REmap 2030: A Renewable Energy Roadmap." Abu Dhabi: IRENA. www.irena.org/remap.

Mah, Daphne Ngar-yin, Guihua Wang, Kevin Lo, Michael K. H. Leung, Peter Hills, and Alex Y. Lo. 2018. "Barriers and Policy Enablers for Solar Photovoltaics (PV) in Cities: Perspectives of Potential Adopters in Hong Kong." *Renewable and Sustainable Energy Reviews* 92 (September): 921–36. <https://doi.org/10.1016/j.rser.2018.04.041>.

Meijer, L. L. J., J. C. C. M. Huijben, A. van Boxstael, and A. G. L. Romme. 2019. "Barriers and Drivers for Technology Commercialization by SMEs in the Dutch Sustainable Energy Sector." *Renewable and Sustainable Energy Reviews* 112 (September): 114–26. <https://doi.org/10.1016/j.rser.2019.05.050>.

Rai, Varun, D. Cale Reeves, and Robert Margolis. 2016. "Overcoming Barriers and Uncertainties in the Adoption of Residential Solar PV." *Renewable Energy* 89 (April): 498–505. <https://doi.org/10.1016/j.renene.2015.11.080>.

Tong, Dan, David J. Farnham, Lei Duan, Qiang Zhang, Nathan S. Lewis, Ken Caldeira, and Steven J. Davis. 2021. "Geophysical Constraints on the Reliability of Solar and Wind Power Worldwide." *Nature Communications* 12 (1): 6146. <https://doi.org/10.1038/s41467-021-26355-z>.

Wang, Mudan, Xianqiang Mao, Youkai Xing, Jianhong Lu, Peng Song, Zhengyan Liu, Zhi Guo, Kevin Tu, and Eric Zusman. 2021. "Breaking down Barriers on PV Trade Will Facilitate Global Carbon Mitigation." *Nature Communications* 12 (1): 6820. <https://doi.org/10.1038/s41467-021-26547-7>.

Xue, Yan, Carmel Margaret Lindkvist, and Alenka Temeljotov-Salaj. 2021. "Barriers and Potential Solutions to the Diffusion of Solar Photovoltaics from the Public-Private-People Partnership Perspective – Case Study of Norway." *Renewable and Sustainable Energy Reviews* 137 (March): 110636. <https://doi.org/10.1016/j.rser.2020.110636>.

Reviewers' Comments:

Reviewer #1:

Remarks to the Author:

I'm broadly happy with the revised manuscript. Some of the authors' responses to my first round of comments could usefully be included in the manuscript (as well as in the responses to me). I've uploaded these suggestions in red, inserted into relevant points of the response to reviewers document.

REVIEWER COMMENTS

Reviewer #1 (Remarks to the Author):

Overall, I think this article is clear, well written, well argued, topical, of consistently high quality, and certainly worthy of publication. My comments are intended mainly as suggestions and thoughts for the authors to consider (but not necessarily feel obliged to act on) in the final version.

Thank you very much for the comments. We agree on the majority of points and have incorporated them into the paper. On the remaining points we have provided justification below'. The paper is now much clearer.

R1: As a general comment, it would be very helpful to either summarise the text that's been added or changed in relation to each of the responses below and/or to upload a track changes version of the manuscript ... either would help reviewers identify more easily how the manuscript has changed.

General Comments

The evidence, approach, and resulting insights in the paper are very similar to Way et al. in Joule (ref 10 but needs updating to published version) albeit with a different systems modelling tool. The main difference in this paper seems to be the discussion of barriers to large-scale RE diffusion, and the sensitisation of results to those barriers (which Way et al. also reflect on to some extent in their discussion) It would be helpful if the authors could explain clearly how and why their work builds on, differs from, or challenges the results of Way et al., and why the different systems modelling approach yields similar (or different) insights.

There are two key differences Way et al and our work. The first is that our model contains the feedback between investment and learning endogenously. The price declines impact our investment decision making core, which further drive price declines. Way et al take a scenario perspective, where investments are chosen exogenously.

The second one is that our model is disaggregated into 70 world regions. As such, we can track how uptake of renewables is likely to evolve in countries with a wide variety of circumstances in terms of technical potential, variability of production for VRE and pricing.

We've added a paragraph detailing how the studies use complementary methodologies to reach a similar conclusion.

Specific Comments

The Word doc manuscript I read had no page or line numbers. I am using the first page with title, abstract, and the beginning of the introduction as p1.

p2 "the rapid disruptive transition" - applies specifically to electricity systems rather than the wider energy system (although clearly these are interlinked). The article generally does a good job of referring specifically to electricity rather than generically to energy, but it would be good if the authors could do this consistently throughout.

Done

p2 “a series of barriers” - misses also the large implied land (or rooftop) requirements for massive solar scale up given its very low energy densities (and ditto wind).

Thank you for the comment. We provide justification below (reviewer 2 report).

p2 “underestimated uptake rates of solar energy” - one of many good examples of a topic covered exhaustively in Way et al. (not referenced here)

Cited in addition to Meng et al

p3 “solar ... high learning rates ... modularity ... replicability” - both Sweerts et al. 2020 and Wilson et al. 20202 show this for diverse sets of energy technologies as a function of unit size and modularity, and these were nicely summarise in IPCC AR6 too

Now cited both Sweerts and Wilson, and removed one of the less relevant citations.

p3 “historical failure ... “ - exogenously imposed diffusion constraints specifically designed to restrain overly rapid technological shifts (to RE) also played a part, particularly in optimisation frameworks

Added and cited to Way et al.

p3 “we use IEA, BNEF ... “ - to calibrate what?

Clarified: “We use IEA data for historical generation, CAPEX and OPEX, BNEF for capacity factors, construction and lifetimes until 2020, IRENA for historical renewables capacity data between 2019 and 2021 and 2021”

R1: Typo repeat of 2021

p3 “long lifetimes prevent ...” - without large-scale premature retirement of capital stock (prior to the end of its technical lifetime) or stranded assets, a topic which would be welcome to revisit in the discussion

Our paper sketches a baseline, a likely scenario when policies are unchanged. It may distract to mention premature retirement of capital stock here too, which usually occurs in scenarios with more explicit phase-outs of fossil-fuels. A more nuanced description of stranded assets can be found in other work with this model (f.i., Semieniuk et al. 2022).

R1: “long lifetimes prevent technological trajectories from changing direction abruptly” ... in the absence of premature retirement of capital stock. As this is already currently happening with coal power in some contexts, I think it’s also worth mentioning in the text – or something note in the text, at least summarising the response given here.

Fig 1 - this is a niggle, but onshore and offshore would benefit from also being wind!

Added

Fig 1 and explanatory text below - there’s no information given on the many other assumptions behind these projections; assumptions which the SSP framework systematically and transparently

varies given baseline uncertainties (e.g., rates of economic growth, demographic change, energy service demand growth, electrification rates to account for electrification of heat and transport, and so on). Can you clarify in brief here, and in detail in Methods, the broader assumptions used in the projections. This absolutely does not have to be linked to the SSP framework, but how the projections position approximately within the SSP framework would then be helpful to compare and contrast results with the large SSP-informed literature.

We included a rough comparison to the SSP framework in the methods section, including information about economic growth and demographic change, and added two references that describe our assumptions in more details.

p5 “learning rate for long-duration storage” - I wasn’t always clear what was being referred to here: is it green H2 for bridging inter-seasonal variation? inter-seasonal or even inter-annual storage is an important issue for high RE grids, so this would merit being spelt out more clearly

Done. We assume hydrogen is used for long-term storage.

Fig 3 - this is a really nicely designed Figure, illustrating a key argument with the viewer in mind ... but would gas not be a better comparator than coal, as gas is dominant, growing, and sets the marginal price in many markets while coal is already in decline in many jurisdictions (+ onshore WIND!)

Thank you for the compliment. Over the last few years, the price of gas has made many large jumps and falls, and it is a bit unclear how this is changing in the future. We fear that adding gas to the figure will lead to some arbitrariness. Changed it to ONSHORE wind.

As this is a simulation study about the future deployment of solar and its relative attractiveness compared to fossil incumbents, this is a weak argument. If the forward-looking simulation of gas LCOE is volatile, what are the exogenous assumptions on which this volatility is based (analogous to wars or pandemics)? What does the 2020-2050 LCOE projection of gas look like? Is it volatile like the recent historical movements? If so, how does the technology selection of electricity generation at the margins in the forward-looking simulations get affected? At the very least, please can you include the gas LCOE bands in S.I. as a comparable figure, and explain justification in the text.

p6 “72% of simulations” - clarify these are the full set of simulations used for robustness testing

Done. These are the full set.

p6 “rules out a subset of SSPs” - these need explaining (for the uninitiated reader) + see my earlier points on it being hard to interpret this comment as no equivalent information to SSP storylines is given for your projections (other than on technology costs)

We have included a discussion on how the model compares with SSP2 in the methods section, and explained what SSPs are in the main text, and what type of SSPs we rule out (the ones with increasing coal shares, such as SSP5).

p7 “suggests that further climate policy ... “ - I think this is a key and important insight from the analysis, and is rightly emphasised again in the discussion ... but it’s corollary is that any and all

remaining price support for RE can be stopped without causing a meaningful dent in uptake rates: is this borne out in jurisdictions that have done this? and does it still hold if cost-related climate policy support for RE is shifted to other large-scale low-carbon options like CCS or nuclear?

We have not researched if remaining price support for RE can be stopped, but there are examples of countries phasing out monetary support for renewables without it affecting uptake (for instance, offshore wind in the Netherlands and Germany (Rentier et al., 2023)).

Countries with small solar and wind industries would likely benefit from keeping subsidies to speed up deployment. Furthermore, alternative RE technologies to wind and solar PV (such as CSP, geothermal and small hydro) are often not attractive to investors, and may play an important role in grid stabilisation, so subsidies remain important for those technologies. The difference in LCOE between nuclear and coal+CCS and solar is quite big. It may be more likely that these will replace fossil fuels plants without CCS if their support is increased, given a similar role they play in the grid.

p8 “notably, new power market rules can be designed ...” - I didn’t understand this point, can you exemplify or explain what generators are that diversify intermittency

Changed “that diversify grid sources of intermittency” with “that complement solar production on a daily to seasonal scale”

p8 “charged directly to consumers ...” - I think more generally there are approaches that require individual generators to pay for intermittency risk one way or the other, versus approaches that collectivise or socialise intermittency risk (through bills, utility charges, regulated requirements, etc.)

Agreed. We investigate a variety of payment options in a separate paper (in preparation) and consider it further discussion out of scope for this paper.

Fig 6 - why is there a strong 2025-2030 peak, what explains this?

The 2025-2035 peak coincides with the fastest-growing part of the S-curve. This is a combination of two factors: firstly, we have only recently reached the point that renewables are cheaper than alternatives, so that investing in renewables makes economic sense. But, the industry in many countries is still growing, and funds there are limits to how fast knowledge and technology can build out, so the peak only happens a few years into the future.

I don’t fully understand this explanation as the inflection point of the S-curve is halfway to saturation and it doesn’t seem realistic that in some jurisdictions this is reached before 2030 (e.g., China), particularly given your second point about industries being immature in many countries. Either way, could you clarify the explanation to the reader of your article too? E.g., in Fig 6 legend. There’s also a typo in your response: “funds”.

Fig 6 legend - “moderate growth worldwide” looks like a tripling in 10 years ... is this moderate??

No, not really, good point. Changed to fast.

p10 “utility-scale requires land ... may be scarce near population centres” - see earlier point, this is quite a throwaway mention of a potentially important constraint. Can you justify your confidence this won’t be a constraint (e.g., in reference to spatially-explicit RE diffusion studies with equivalent TW deployments to your projections?)

We agree we wrote that too confidently, and have now dedicated a paragraph to the issue. We believe it is less of a physical barrier, but rather a political barrier, as the technical potential is available in a large majority of regions. Our model incorporates a maximum technical potential, using the sum of residential technical potential and utility-scale technical potential. These numbers are derived from sources with an explicit spatial component. We are disregarding the technical potential of “floatovoltaics” (<https://www.nature.com/articles/d41586-022-01525-1>), which can fully provide solar demand in some countries, but is a newer application.

There’s a typo in this new paragraph, with a stray “may is”. It’s also based on one reference (53) which is for the US not global.

p10 conclusion - first para echoes Way et al., see general comment about explaining value added of your study

We’ve included more references to Way et al, and add a paragraph that explains how, with quite different methodology, we reach similar conclusion.

p10 “issues that could hinder ...” - as well as land, is there not relevant insight into northern versus southern latitudes and output per installed capacity under different insolation regimes? is this partly why Russia seems the laggard region (relatively speaking)

On a global scale, we do not see this as a major barrier. It is true of course that solar is less attractive in Northern latitudes, and our model has limited uptake of solar in Scandinavian and Baltic countries. But already in countries as far north as Germany, we model a significant role for solar. Southern Russia has a good solar potential, but still very little solar production. The abundance of cheap nuclear and fossil fuels is a larger barrier there than insolation per se.

1 “GENIE ... soft-coupled; affected by the global economy ... “ - weird grammar so a bit hard to follow: basically emissions from FTT-E3ME drive climate model, but climate impacts do not feedback into FTT-E3ME + ref for Mercure et al. missing

Reworded and citation added.

p12 “similar seeding is performed for CSP ... “ - is this because a technology with zero deployment will never be selected? please clarify

Indeed, clarified

p12 “long-term storage a more modest learning ... “ - see earlier point, please be clearer about what this long-term storage is

Clarified. It is assumed to be hydrogen.

p13 E3ME - how are static IO tables in E3ME consistent with dynamic representations of technological substitution in FTT? (or are the IO tables dynamically coupled into the FTT model in some way)

The IO tables are dynamically coupled to the FTT models via the energy balances. In specific, the coefficients for coal, oil and gas, manufactured fuels, electricity and "gas, steam & air

conditioning” are adjusted based on the coupling with the FTT models. I’ve added this to the methods section.

p14 GENIE - I couldn’t see how GENIE was used in this study as there’s no climate results reported ... it would actually be interesting to see what the warming outcome is of the reference projections, relative to the scenario literature

I have removed the paragraph on GENIE. We do indeed not report on outcomes from GENIE in this paper. I have chosen not to include a temperature projection, as some key parts of the model (heat/transport) are undergoing major improvements, so this number would possibly not be very realistic.

References - need a good cleaning and checking: e.g., refs 18 and 21 are the same; are refs 12 and 34 also the same?

18 and 21 now separated. Refs 12 and 34 were by the same author, but not the same.

References review

Sweerts, B., R. J. Detz and B. van der Zwaan (2020). "Evaluating the Role of Unit Size in Learning-by-Doing of Energy Technologies." *Joule* 4(5): 967-970. DOI: 10.1016/j.joule.2020.03.010

Way, R., M. C. Ives, P. Mealy and J. D. Farmer (2022). "Empirically grounded technology forecasts and the energy transition." *Joule* 6(9): 2057-2082. DOI: <https://doi.org/10.1016/j.joule.2022.08.009>

Wilson, C., A. Grubler, N. Bento, S. Healey, S. De Stercke and C. Zimm (2020). "Granular technologies to accelerate decarbonization." *Science* 368(6486): 36-39. DOI: 10.1126/science.aaz8060

References in answer:

Rentier G, Lelieveldt H, Kramer G (2023) Institutional constellations and policy instruments for offshore wind power around the North sea. *Energy Policy* 173

Semieniuk, Gregor, Philip B. Holden, Jean-Francois Mercure, Pablo Salas, Hector Pollitt, Katharine Jobson, Pim Vercoulen, Unnada Chewpreecha, Neil R. Edwards, and Jorge E. Viñuales. 2022. "Stranded Fossil-Fuel Assets Translate to Major Losses for Investors in Advanced Economies." *Nature Climate Change* 12 (6): 532–38. <https://doi.org/10.1038/s41558-022-01356-y>.

Reviewer #2 (Remarks to the Author):

Thank you for giving me the opportunity to read this paper. I believe the paper makes an important contribution to the field of renewable energy adoption, using the lens of learning and diffusion models. I have several suggestions for improvements:

Thank you for the detailed review comments. We agree with most comments and have incorporated them into the paper. On the remaining points we have provided an explanation of our reasoning below. The paper is now much clearer.

The authors should explain how they have identified and selected the four barriers or bottlenecks to diffusion of renewable/solar energy, including grid resilience, investment barriers, supply chains, and political economy, and why not others.

Thank you for the comment. Different barriers, bottlenecks and problems affect different renewable energy depending on the source, geographical contexts, country-specific market structures and levels of maturity (IRENA, 2019). We reviewed the literature and selected the four *worldwide, high level barriers* that builds upon the four levels (technological, economic, policy and socio-political) flagged in IRENA, 2019

Scholars have studied context-specific barriers that hamper successful scale-up of solar PV (Table below) focusing on non-technical (*soft*) barriers such as perceptions, information, access to capital and uncertainty (Meijer et al. 2019; Mah et al. 2018; Xue, Lindkvist, and Temeljotov-Salaj 2021; Rai, Reeves, and Margolis 2016; Do et al. 2020)) and on technical factors, including grid resilience, financing mobilisation, trade and geophysical constraints (IRENA 2014; Tong et al. 2021; Wang et al. 2021). Our four hampering factors (grid resilience, availability of finance, supply chain and the pace of the transition) interest and span across all identified and literature-informed barriers. Moreover, they build upon and expand the four core limitations described in IRENA (2019): technological, policy, market and economic and regulatory, political and social barriers.

Grid resilience encompasses technical, but also political and regulatory issues as policy mixes can overcome intermittency. Finance mobilisation interests economic, social and political barriers as a more equal distribution of financial resources will expand solar PV in emerging economies too. Supply chain barriers lead to geopolitical, policy and technological tensions and frictions as the critical mineral industry and technology providers in the renewable energy space will have to align and coordinate to satisfy future demand. Finally, the pace of the transition involves all four IRENA (2019) building blocks.

The authors could clarify whether their calculations are based on residential solar PV or utility-scale solar PV. The LCOE will be different for these two types of solar PV.

Clarified in methods (we use utility-scale pricing). In future work we plan to incorporate the higher residential LCOE and avoided taxes for consumers from producing their own electricity.

A discussion could be added on the role of concentrated solar power. Studies show that solar PV is deployed earlier due to sharp price decreases in the last decades and that concentrated solar

thermal technology will gain a competitive advantage when PV has a high market share and its system integration costs become too high (e.g., Pietzcker et al., 2014). What role could concentrated solar power play in grid resilience?

In our model, CSP plays a growing, but moderate role. We include the advantage of CSP as a dispatchable source, and only require it to pay for seasonal storage (as thermal storage with most CSP system precludes the need for short-term Lithium-ion storage). However, as we are investigating a scenario without additional policy, we may not fully capture the advantages of CSP: in a scenario with a fossil fuel phase-out, the role of CSP could become much larger. Small technologies usually do not replace established technologies without a targeted push, unless the advantages far outweigh the system's inertia. Since Pietzcker et al, the relative advantage of CSP versus solar PV has slightly diminished, as lithium-ion batteries have seen a tremendous drop in prices. I've added a short description to the results:

Concentrated solar power grows over the entire period, but without targeted policy its overall share in the power mix remains small, despite its advantage as a dispatchable source of electricity.

Our methodology relies strongly on historical trends, and in the last few years CSP has had little growth, and even a decline in 2021 (REN21, 2022).

The authors state that: “no mechanism guarantees that optimal grids are achieved if left to market forces, especially in contexts of diverging technology costs, and solar dominance could become self-limiting.” The authors could consider the fact that renewable energy companies will have an incentive to invest in the grid (and they do invest in grid and storage technology) to enable their industry to overcome barriers to solar and wind adoption.

This is certainly true, but the extent to which they can make these investments depends on the market structure, which does not automatically align with a theoretical optimality. Market decisions are often made on short time scales. For instance, a grid dominated by solar would benefit from the presence of pumped hydro. However, the economics of pumped hydro are difficult in the current situation, and, in combination with market uncertainty, the long construction times would not allow for these plants to be built at an optimal time (MIT, 2022).

The authors also state that: “our model suggests that the allocation of storage costs to the grid and charged directly to consumers incentivises more renewables diffusion than requiring each project to provide their own storage”. Regarding this statement I wonder what this suggestion does to the learning curves considered in the models and the LCOE that include system storage costs. Private RE actors who compete for solar projects innovate and invest in grid and storage technology and through competition drive down costs. Is the learning curve and the cost reduction faster for private actors than when we leave the investments up to (often) publicly owned national or regional monopolies for grid operation?

That's a very interesting question. In our model, learning curves are dependent on the cumulative deployment of a technology, independent on who deploys. I'm unaware of research that directly answer that question.

Winfield et al. discussed this question from a multi-level perspective approach¹. They concluded that liberalised and monopoly markets have different opportunities to go from niche-level innovation to a regime transition, and thereby to bring down costs. Liberalised markets will have niches created by a diversity of third-party investors. Monopolies can choose to create shielded niches for innovation to take place. The paper did not conclude that either option is better.

In barrier 2, does the suggestion on lines 241-243 refer to involvement of the World bank and other investment funds for developing countries, i.e., those institutions that European-based solar project developers already collaborate with for their solar projects in developing economies?

The World Bank has a relatively small balance sheet relative to the size of the problem, so while they play an important role, the overall scale of the challenge is much larger and should include the participation of a wide scale of financial institutions.

The conclusion could highlight the novelty of the results of this paper that result from using the simulation methods.

We have accentuated the difference from Way et al (who use statistical methodology) in the main text, and clarified the how the simulation method with “decision-making by investors” differs from a “social planner” optimisation method in the methodology section.

Minor comments:

The sentence on lines 96-98 requires a source, or an explanation as to why these are main reasons for the failure of existing models.

Provided citations to Way et al and Jaxa-Rosen et al, and amended text to remove “optimisation models” and add “imposition of maximum deployment levels” per these sources.

The sentence in line 130 requires more explanation; it is not clear how this follows from the above evidence or arguments.

Moved the sentence to after the uncertainty analysis, which support the sentence better.

The sentence in lines 133-135 seems to contradict itself; if the slow uptake is attributed to non-pecuniary aspects how would lower prices help?

Clarified: when prices fall sufficiently far under alternatives, the pecuniary aspects overcome the nonpecuniary aspects.

Figure 1 could refer to offshore wind and onshore wind. Figure 4b needs to define VRE and a legend is needed to explain the numbers on the x-axis of figure 4e.

What is the source of figure 6? Number 42?

Done. Figure 6 is based on model output, now clarified.

¹ Winfield, Shokrzadeh, and Jones, ‘Energy Policy Regime Change and Advanced Energy Storage’.

Citations review

Pietzcker, R., Stetter, D., Manger, S., and Luderer, G. (2014). Using the sun to decarbonize the power sector: The economic potential of photovoltaics and concentrating solar power. *Applied Energy*, 135, 704-720.

Citations answers:

REN21. 2022. Renewables 2022 Global Status Report. Paris: REN21 Secretariat. ISBN 978-3-948393-04-5

IRENA (2019), *Global energy transformation: A roadmap to 2050* (2019 edition)

Do, Thang Nam, Paul J. Burke, Kenneth G. H. Baldwin, and Chinh The Nguyen. 2020. "Underlying Drivers and Barriers for Solar Photovoltaics Diffusion: The Case of Vietnam." *Energy Policy* 144 (September): 111561. <https://doi.org/10.1016/j.enpol.2020.111561>.

IRENA. 2014. "REmap 2030: A Renewable Energy Roadmap." Abu Dhabi: IRENA. www.irena.org/remap.

Mah, Daphne Ngar-yin, Guihua Wang, Kevin Lo, Michael K. H. Leung, Peter Hills, and Alex Y. Lo. 2018. "Barriers and Policy Enablers for Solar Photovoltaics (PV) in Cities: Perspectives of Potential Adopters in Hong Kong." *Renewable and Sustainable Energy Reviews* 92 (September): 921–36. <https://doi.org/10.1016/j.rser.2018.04.041>.

Meijer, L. L. J., J. C. C. M. Huijben, A. van Boxtael, and A. G. L. Romme. 2019. "Barriers and Drivers for Technology Commercialization by SMEs in the Dutch Sustainable Energy Sector." *Renewable and Sustainable Energy Reviews* 112 (September): 114–26. <https://doi.org/10.1016/j.rser.2019.05.050>.

Rai, Varun, D. Cale Reeves, and Robert Margolis. 2016. "Overcoming Barriers and Uncertainties in the Adoption of Residential Solar PV." *Renewable Energy* 89 (April): 498–505. <https://doi.org/10.1016/j.renene.2015.11.080>.

Tong, Dan, David J. Farnham, Lei Duan, Qiang Zhang, Nathan S. Lewis, Ken Caldeira, and Steven J. Davis. 2021. "Geophysical Constraints on the Reliability of Solar and Wind Power Worldwide." *Nature Communications* 12 (1): 6146. <https://doi.org/10.1038/s41467-021-26355-z>.

Wang, Mudan, Xianqiang Mao, Youkai Xing, Jianhong Lu, Peng Song, Zhengyan Liu, Zhi Guo, Kevin Tu, and Eric Zusman. 2021. "Breaking down Barriers on PV Trade Will Facilitate Global Carbon Mitigation." *Nature Communications* 12 (1): 6820. <https://doi.org/10.1038/s41467-021-26547-7>.

Xue, Yan, Carmel Margaret Lindkvist, and Alenka Temeljotov-Salaj. 2021. "Barriers and Potential Solutions to the Diffusion of Solar Photovoltaics from the Public-Private-People Partnership Perspective – Case Study of Norway." *Renewable and Sustainable Energy Reviews* 137 (March): 110636. <https://doi.org/10.1016/j.rser.2020.110636>.

Reviewer #2:

Remarks to the Author:

I have read the revised version of the manuscript and the response to my earlier comments. I believe the authors have sufficiently addressed my suggestions, and believe the paper makes an important contribution to research on solar energy, innovation diffusion and transitions to low-carbon sectors.

REVIEWER COMMENTS

Reviewer #1 (Remarks to the Author):

Overall, I think this article is clear, well written, well argued, topical, of consistently high quality, and certainly worthy of publication. My comments are intended mainly as suggestions and thoughts for the authors to consider (but not necessarily feel obliged to act on) in the final version.

Thank you very much for the comments. We agree on the majority of points and have incorporated them into the paper. On the remaining points we have provided justification below'. The paper is now much clearer.

R1: As a general comment, it would be very helpful to either summarise the text that's been added or changed in relation to each of the responses below and/or to upload a track changes version of the manuscript ... either would help reviewers identify more easily how the manuscript has changed.

Thanks for the additional feedback! Apologies I forgot to upload the tracked changes document. Have done so for the latest changes.

General Comments

The evidence, approach, and resulting insights in the paper are very similar to Way et al. in Joule (ref 10 but needs updating to published version) albeit with a different systems modelling tool. The main difference in this paper seems to be the discussion of barriers to large-scale RE diffusion, and the sensitisation of results to those barriers (which Way et al. also reflect on to some extent in their discussion) It would be helpful if the authors could explain clearly how and why their work builds on, differs from, or challenges the results of Way et al., and why the different systems modelling approach yields similar (or different) insights.

There are two key differences Way et al and our work. The first is that our model contains the feedback between investment and learning endogenously. The price declines impact our investment decision making core, which further drive price declines. Way et al take a scenario perspective, where investments are chosen exogenously.

The second one is that our model is disaggregated into 70 world regions. As such, we can track how uptake of renewables is likely to evolve in countries with a wide variety of circumstances in terms of technical potential, variability of production for VRE and pricing.

We've added a paragraph detailing how the studies use complementary methodologies to reach a similar conclusion.

Specific Comments

The Word doc manuscript I read had no page or line numbers. I am using the first page with title, abstract, and the beginning of the introduction as p1.

p2 "the rapid disruptive transition" - applies specifically to electricity systems rather than the wider energy system (although clearly these are interlinked). The article generally does a good job of referring specifically to electricity rather than generically to energy, but it would be good if the authors could do this consistently throughout.

Done

p2 “a series of barriers” - misses also the large implied land (or rooftop) requirements for massive solar scale up given its very low energy densities (and ditto wind).

Thank you for the comment. We provide justification below (reviewer 2 report).

p2 “underestimated uptake rates of solar energy” - one of many good examples of a topic covered exhaustively in Way et al. (not referenced here)

Cited in addition to Meng et al

p3 “solar ... high learning rates ... modularity ... replicability” - both Sweerts et al. 2020 and Wilson et al. 2020 show this for diverse sets of energy technologies as a function of unit size and modularity, and these were nicely summarised in IPCC AR6 too

Now cited both Sweerts and Wilson, and removed one of the less relevant citations.

p3 “historical failure ... “ - exogenously imposed diffusion constraints specifically designed to restrain overly rapid technological shifts (to RE) also played a part, particularly in optimisation frameworks

Added and cited to Way et al.

p3 “we use IEA, BNEF ... “ - to calibrate what?

Clarified: “We use IEA data for historical generation, CAPEX and OPEX, BNEF for capacity factors, construction and lifetimes until 2020, IRENA for historical renewables capacity data between 2019 and 2021 and 2021”

R1: Typo repeat of 2021

Corrected, apologies.

p3 “long lifetimes prevent ...” - without large-scale premature retirement of capital stock (prior to the end of its technical lifetime) or stranded assets, a topic which would be welcome to revisit in the discussion

Our paper sketches a baseline, a likely scenario when policies are unchanged. It may distract to mention premature retirement of capital stock here too, which usually occurs in scenarios with more explicit phase-outs of fossil-fuels. A more nuanced description of stranded assets can be found in other work with this model (f.i., Semieniuk et al. 2022).

R1: “long lifetimes prevent technological trajectories from changing direction abruptly” ... in the absence of premature retirement of capital stock. As this is already currently happening with coal power in some contexts, I think it’s also worth mentioning in the text – or something note in the text, at least summarising the response given here.

We’ve added the following to the methodology section:

As a conservative assumption, we do not include a premature retirement of power plants when their marginal costs rise above the LCOE of newly installed power plants. We also do not include the possibility to extend the lifetimes of power plants.

Fig 1 - this is a niggle, but onshore and offshore would benefit from also being wind!

Added

Fig 1 and explanatory text below - there's no information given on the many other assumptions behind these projections; assumptions which the SSP framework systematically and transparently varies given baseline uncertainties (e.g., rates of economic growth, demographic change, energy service demand growth, electrification rates to account for electrification of heat and transport, and so on). Can you clarify in brief here, and in detail in Methods, the broader assumptions used in the projections. This absolutely does not have to be linked to the SSP framework, but how the projections position approximately within the SSP framework would then be helpful to compare and contrast results with the large SSP-informed literature.

We included a rough comparison to the SSP framework in the methods section, including information about economic growth and demographic change, and added two references that describe our assumptions in more details.

p5 "learning rate for long-duration storage" - I wasn't always clear what was being referred to here: is it green H2 for bridging inter-seasonal variation? inter-seasonal or even inter-annual storage is an important issue for high RE grids, so this would merit being spelt out more clearly

Done. We assume hydrogen is used for long-term storage.

Fig 3 - this is a really nicely designed Figure, illustrating a key argument with the viewer in mind ... but would gas not be a better comparator than coal, as gas is dominant, growing, and sets the marginal price in many markets while coal is already in decline in many jurisdictions (+ onshore WIND!)

Thank you for the compliment. Over the last few years, the price of gas has made many large jumps and falls, and it is a bit unclear how this is changing in the future. We fear that adding gas to the figure will lead to some arbitrariness. Changed it to ONSHORE wind.

As this is a simulation study about the future deployment of solar and its relative attractiveness compared to fossil incumbents, this is a weak argument. If the forward-looking simulation of gas LCOE is volatile, what are the exogenous assumptions on which this volatility is based (analogous to wars or pandemics)? What does the 2020-2050 LCOE projection of gas look like? Is it volatile like the recent historical movements? If so, how does the technology selection of electricity generation at the margins in the forward-looking simulations get affected? At the very least, please can you include the gas LCOE bands in S.I. as a comparable figure, and explain justification in the text.

We've now included gas in the figure in the main paper. We do not capture the full price volatility of gas in our model (at the moment, we only capture volatility from a short period of time), and certainly not the spike that was caused by the recent events or wars in general. I've added this to the methodology section: "Volatility makes the standard deviation of LCOE larger, which causes a slower take-up of the technology in regions where gas is competitive."

p6 “72% of simulations” - clarify these are the full set of simulations used for robustness testing

Done. These are the full set.

p6 “rules out a subset of SSPs” - these need explaining (for the uninitiated reader) + see my earlier points on it being hard to interpret this comment as no equivalent information to SSP storylines is given for your projections (other than on technology costs)

We have included a discussion on how the model compares with SSP2 in the methods section, and explained what SSPs are in the main text, and what type of SSPs we rule out (the ones with increasing coal shares, such as SSP5).

p7 “suggests that further climate policy ... “ - I think this is a key and important insight from the analysis, and is rightly emphasised again in the discussion ... but it’s corollary is that any and all remaining price support for RE can be stopped without causing a meaningful dent in uptake rates: is this borne out in jurisdictions that have done this? and does it still hold if cost-related climate policy support for RE is shifted to other large-scale low-carbon options like CCS or nuclear?

We have not researched if remaining price support for RE can be stopped, but there are examples of countries phasing out monetary support for renewables without it affecting uptake (for instance, offshore wind in the Netherlands and Germany (Rentier et al., 2023)).

Countries with small solar and wind industries would likely benefit from keeping subsidies to speed up deployment. Furthermore, alternative RE technologies to wind and solar PV (such as CSP, geothermal and small hydro) are often not attractive to investors, and may play an important role in grid stabilisation, so subsidies remain important for those technologies. The difference in LCOE between nuclear and coal+CCS and solar is quite big. It may be more likely that these will replace fossil fuels plants without CCS if their support is increased, given a similar role they play in the grid.

p8 “notably, new power market rules can be designed ...” - I didn’t understand this point, can you exemplify or explain what generators are that diversify intermittency

Changed “that diversify grid sources of intermittency” with “that complement solar production on a daily to seasonal scale”

p8 “charged directly to consumers ...” - I think more generally there are approaches that require individual generators to pay for intermittency risk one way or the other, versus approaches that collectivise or socialise intermittency risk (through bills, utility charges, regulated requirements, etc.)

Agreed. We investigate a variety of payment options in a separate paper (in preparation) and consider it further discussion out of scope for this paper.

Fig 6 - why is there a strong 2025-2030 peak, what explains this?

The 2025-2035 peak coincides with the fastest-growing part of the S-curve. This is a combination of two factors: firstly, we have only recently reached the point that renewables are cheaper than alternatives, so that investing in renewables makes economic sense. But, the industry in many countries is still growing, and funds there are limits to how fast knowledge and technology can build out, so the peak only happens a few years into the future.

I don't fully understand this explanation as the inflection point of the S-curve is halfway to saturation and it doesn't seem realistic that in some jurisdictions this is reached before 2030 (e.g., China), particularly given your second point about industries being immature in many countries. Either way, could you clarify the explanation to the reader of your article too? E.g., in Fig 6 legend. There's also a typo in your response: "funds".

While the industry is still growing, I would not call it immature. In our model, we are close to the inflection point in China, and by 2028 new additions do not go up anymore. I only explained the start of the peak in my previous answer, not why it is going down again, while new additions remain approximately constant. I've added "The strong peak around 2030 for China and India is explained by a saturation in addition of additional solar capacity, in combination with a growing GDP and declining solar costs" to the figure caption.

Fig 6 legend - "moderate growth worldwide" looks like a tripling in 10 years ... is this moderate??

No, not really, good point. Changed to fast.

p10 "utility-scale requires land ... may be scarce near population centres" - see earlier point, this is quite a throwaway mention of a potentially important constraint. Can you justify your confidence this won't be a constraint (e.g., in reference to spatially-explicit RE diffusion studies with equivalent TW deployments to your projections?)

We agree we wrote that too confidently, and have now dedicated a paragraph to the issue. We believe it is less of a physical barrier, but rather a political barrier, as the technical potential is available in a large majority of regions. Our model incorporates a maximum technical potential, using the sum of residential technical potential and utility-scale technical potential. These numbers are derived from sources with an explicit spatial component. We are disregarding the technical potential of "floatovoltaics" (<https://www.nature.com/articles/d41586-022-01525-1>), which can fully provide solar demand in some countries, but is a newer application.

There's a typo in this new paragraph, with a stray "may is". It's also based on one reference (53) which is for the US not global.

Corrected the typo and have added an additional reference for India (Deshmukh et al, 2019).

p10 conclusion - first para echoes Way et al., see general comment about explaining value added of your study

We've included more references to Way et al, and add a paragraph that explains how, with quite different methodology, we reach similar conclusion.

p10 "issues that could hinder ..." - as well as land, is there not relevant insight into northern versus southern latitudes and output per installed capacity under different insolation regimes? is this partly why Russia seems the laggard region (relatively speaking)

On a global scale, we do not see this as a major barrier. It is true of course that solar is less attractive in Northern latitudes, and our model has limited uptake of solar in Scandinavian and Baltic countries. But already in countries as far north as Germany, we model a significant role for solar. Southern Russia has a good solar potential, but still very little solar production. The abundance of cheap nuclear and fossil fuels is a larger barrier there than insolation per se.

1 "GENIE ... soft-coupled; affected by the global economy ... " - weird grammar so a bit hard to follow: basically emissions from FTT-E3ME drive climate model, but climate impacts do not feedback into FTT-E3ME + ref for Mercure et al. missing

Reworded and citation added.

p12 "similar seeding is performed for CSP ... " - is this because a technology with zero deployment will never be selected? please clarify

Indeed, clarified

p12 "long-term storage a more modest learning ... " - see earlier point, please be clearer about what this long-term storage is

Clarified. It is assumed to be hydrogen.

p13 E3ME - how are static IO tables in E3ME consistent with dynamic representations of technological substitution in FTT? (or are the IO tables dynamically coupled into the FTT model in some way)

The IO tables are dynamically coupled to the FTT models via the energy balances. In specific, the coefficients for coal, oil and gas, manufactured fuels, electricity and "gas, steam & air conditioning" are adjusted based on the coupling with the FTT models. I've added this to the methods section.

p14 GENIE - I couldn't see how GENIE was used in this study as there's no climate results reported ... it would actually be interesting to see what the warming outcome is of the reference projections, relative to the scenario literature

I have removed the paragraph on GENIE. We do indeed not report on outcomes from GENIE in this paper. I have chosen not to include a temperature projection, as some key parts of the model (heat/transport) are undergoing major improvements, so this number would possibly not be very realistic.

References - need a good cleaning and checking: e.g., refs 18 and 21 are the same; are refs 12 and 34 also the same?

18 and 21 now separated. Refs 12 and 34 were by the same author, but not the same.

References review

Sweerts, B., R. J. Detz and B. van der Zwaan (2020). "Evaluating the Role of Unit Size in Learning-by-Doing of Energy Technologies." *Joule* 4(5): 967-970. DOI: [10.1016/j.joule.2020.03.010](https://doi.org/10.1016/j.joule.2020.03.010)

Way, R., M. C. Ives, P. Mealy and J. D. Farmer (2022). "Empirically grounded technology forecasts and the energy transition." *Joule* 6(9): 2057-2082. DOI: <https://doi.org/10.1016/j.joule.2022.08.009>

Wilson, C., A. Grubler, N. Bento, S. Healey, S. De Stercke and C. Zimm (2020). "Granular technologies to accelerate decarbonization." *Science* 368(6486): 36-39. DOI: [10.1126/science.aaz8060](https://doi.org/10.1126/science.aaz8060)

References in answer:

Rentier G, Lelieveldt H, Kramer G (2023) Institutional constellations and policy instruments for offshore wind power around the North sea. *Energy Policy* 173

Semieniuk, Gregor, Philip B. Holden, Jean-Francois Mercure, Pablo Salas, Hector Pollitt, Katharine Jobson, Pim Vercoulen, Unnada Chewpreecha, Neil R. Edwards, and Jorge E. Viñuales. 2022. "Stranded Fossil-Fuel Assets Translate to Major Losses for Investors in Advanced Economies." *Nature Climate Change* 12 (6): 532–38. <https://doi.org/10.1038/s41558-022-01356-y>.

Additional reference in second review round:

Deshmukh, Ranjit, Grace C. Wu, Duncan S. Callaway, and Amol Phadke. 2019. "Geospatial and Techno-Economic Analysis of Wind and Solar Resources in India." *Renewable Energy* 134 (April): 947–60. <https://doi.org/10.1016/j.renene.2018.11.073>.

Reviewer #2 (Remarks to the Author):

Thank you for giving me the opportunity to read this paper. I believe the paper makes an important contribution to the field of renewable energy adoption, using the lens of learning and diffusion models. I have several suggestions for improvements:

Thank you for the detailed review comments. We agree with most comments and have incorporated them into the paper. On the remaining points we have provided an explanation of our reasoning below. The paper is now much clearer.

The authors should explain how they have identified and selected the four barriers or bottlenecks to diffusion of renewable/solar energy, including grid resilience, investment barriers, supply chains, and political economy, and why not others.

Thank you for the comment. Different barriers, bottlenecks and problems affect different renewable energy depending on the source, geographical contexts, country-specific market structures and levels of maturity (IRENA, 2019). We reviewed the literature and selected the four worldwide, high level barriers that builds upon the four levels (technological, economic, policy and socio-political) flagged in IRENA, 2019

Scholars have studied context-specific barriers that hamper successful scale-up of solar PV (Table below) focusing on non-technical (*soft*) barriers such as perceptions, information, access to capital and uncertainty (Meijer et al. 2019; Mah et al. 2018; Xue, Lindkvist, and Temeljotov-Salaj 2021; Rai, Reeves, and Margolis 2016; Do et al. 2020)) and on technical factors, including grid resilience, financing mobilisation, trade and geophysical constraints (IRENA 2014; Tong et al. 2021; Wang et al. 2021). Our four hampering factors (grid resilience, availability of finance, supply chain and the pace of the transition) interest and span across all identified and literature-informed barriers. Moreover, they build upon and expand the four core limitations described in IRENA (2019): technological, policy, market and economic and regulatory, political and social barriers.

Grid resilience encompasses technical, but also political and regulatory issues as policy mixes can overcome intermittency. Finance mobilisation interests economic, social and political barriers as a more equal distribution of financial resources will expand solar PV in emerging economies too. Supply chain barriers lead to geopolitical, policy and technological tensions and frictions as the critical mineral industry and technology providers in the renewable energy space will have to align and coordinate to satisfy future demand. Finally, the pace of the transition involves all four IRENA (2019) building blocks.

The authors could clarify whether their calculations are based on residential solar PV or utility-scale solar PV. The LCOE will be different for these two types of solar PV.

Clarified in methods (we use utility-scale pricing). In future work we plan to incorporate the higher residential LCOE and avoided taxes for consumers from producing their own electricity.

A discussion could be added on the role of concentrated solar power. Studies show that solar PV is deployed earlier due to sharp price decreases in the last decades and that concentrated solar

thermal technology will gain a competitive advantage when PV has a high market share and its system integration costs become too high (e.g., Pietzcker et al., 2014). What role could concentrated solar power play in grid resilience?

In our model, CSP plays a growing, but moderate role. We include the advantage of CSP as a dispatchable source, and only require it to pay for seasonal storage (as thermal storage with most CSP system precludes the need for short-term Lithium-ion storage). However, as we are investigating a scenario without additional policy, we may not fully capture the advantages of CSP: in a scenario with a fossil fuel phase-out, the role of CSP could become much larger. Small technologies usually do not replace established technologies without a targeted push, unless the advantages far outweigh the system's inertia. Since Pietzcker et al, the relative advantage of CSP versus solar PV has slightly diminished, as lithium-ion batteries have seen a tremendous drop in prices. I've added a short description to the results:

Concentrated solar power grows over the entire period, but without targeted policy its overall share in the power mix remains small, despite its advantage as a dispatchable source of electricity.

Our methodology relies strongly on historical trends, and in the last few years CSP has had little growth, and even a decline in 2021 (REN21, 2022).

The authors state that: "no mechanism guarantees that optimal grids are achieved if left to market forces, especially in contexts of diverging technology costs, and solar dominance could become self-limiting." The authors could consider the fact that renewable energy companies will have an incentive to invest in the grid (and they do invest in grid and storage technology) to enable their industry to overcome barriers to solar and wind adoption.

This is certainly true, but the extent to which they can make these investments depends on the market structure, which does not automatically align with a theoretical optimality. Market decisions are often made on short time scales. For instance, a grid dominated by solar would benefit from the presence of pumped hydro. However, the economics of pumped hydro are difficult in the current situation, and, in combination with market uncertainty, the long construction times would not allow for these plants to be built at an optimal time (MIT, 2022).

The authors also state that: "our model suggests that the allocation of storage costs to the grid and charged directly to consumers incentivises more renewables diffusion than requiring each project to provide their own storage". Regarding this statement I wonder what this suggestion does to the learning curves considered in the models and the LCOE that include system storage costs. Private RE actors who compete for solar projects innovate and invest in grid and storage technology and through competition drive down costs. Is the learning curve and the cost reduction faster for private actors than when we leave the investments up to (often) publicly owned national or regional monopolies for grid operation?

That's a very interesting question. In our model, learning curves are dependent on the cumulative deployment of a technology, independent on who deploys. I'm unaware of research that directly answer that question.

Winfield et al. discussed this question from a multi-level perspective approach¹. They concluded that liberalised and monopoly markets have different opportunities to go from niche-level innovation to a regime transition, and thereby to bring down costs. Liberalised markets will have niches created by a diversity of third-party investors. Monopolies can choose to create shielded niches for innovation to take place. The paper did not conclude that either option is better.

In barrier 2, does the suggestion on lines 241-243 refer to involvement of the World bank and other investment funds for developing countries, i.e., those institutions that European-based solar project developers already collaborate with for their solar projects in developing economies?

The World Bank has a relatively small balance sheet relative to the size of the problem, so while they play an important role, the overall scale of the challenge is much larger and should include the participation of a wide scale of financial institutions.

The conclusion could highlight the novelty of the results of this paper that result from using the simulation methods.

We have accentuated the difference from Way et al (who use statistical methodology) in the main text, and clarified the how the simulation method with “decision-making by investors” differs from a “social planner” optimisation method in the methodology section.

Minor comments:

The sentence on lines 96-98 requires a source, or an explanation as to why these are main reasons for the failure of existing models.

Provided citations to Way et al and Jaxa-Rosen et al, and amended text to remove “optimisation models” and add “imposition of maximum deployment levels” per these sources.

The sentence in line 130 requires more explanation; it is not clear how this follows from the above evidence or arguments.

Moved the sentence to after the uncertainty analysis, which support the sentence better.

The sentence in lines 133-135 seems to contradict itself; if the slow uptake is attributed to non-pecuniary aspects how would lower prices help?

Clarified: when prices fall sufficiently far under alternatives, the pecuniary aspects overcome the nonpecuniary aspects.

Figure 1 could refer to offshore wind and onshore wind. Figure 4b needs to define VRE and a legend is needed to explain the numbers on the x-axis of figure 4e.

What is the source of figure 6? Number 42?

Done. Figure 6 is based on model output, now clarified.

¹ Winfield, Shokrzadeh, and Jones, ‘Energy Policy Regime Change and Advanced Energy Storage’.

Citations review

Pietzcker, R., Stetter, D., Manger, S., and Luderer, G. (2014). Using the sun to decarbonize the power sector: The economic potential of photovoltaics and concentrating solar power. *Applied Energy*, 135, 704-720.

Citations answers:

REN21. 2022. Renewables 2022 Global Status Report. Paris: REN21 Secretariat. ISBN 978-3-948393-04-5

IRENA (2019), *Global energy transformation: A roadmap to 2050* (2019 edition)

Do, Thang Nam, Paul J. Burke, Kenneth G. H. Baldwin, and Chinh The Nguyen. 2020. "Underlying Drivers and Barriers for Solar Photovoltaics Diffusion: The Case of Vietnam." *Energy Policy* 144 (September): 111561. <https://doi.org/10.1016/j.enpol.2020.111561>.

IRENA. 2014. "REmap 2030: A Renewable Energy Roadmap." Abu Dhabi: IRENA. www.irena.org/remap.

Mah, Daphne Ngar-yin, Guihua Wang, Kevin Lo, Michael K. H. Leung, Peter Hills, and Alex Y. Lo. 2018. "Barriers and Policy Enablers for Solar Photovoltaics (PV) in Cities: Perspectives of Potential Adopters in Hong Kong." *Renewable and Sustainable Energy Reviews* 92 (September): 921–36. <https://doi.org/10.1016/j.rser.2018.04.041>.

Meijer, L. L. J., J. C. C. M. Huijben, A. van Boxstael, and A. G. L. Romme. 2019. "Barriers and Drivers for Technology Commercialization by SMEs in the Dutch Sustainable Energy Sector." *Renewable and Sustainable Energy Reviews* 112 (September): 114–26. <https://doi.org/10.1016/j.rser.2019.05.050>.

Rai, Varun, D. Cale Reeves, and Robert Margolis. 2016. "Overcoming Barriers and Uncertainties in the Adoption of Residential Solar PV." *Renewable Energy* 89 (April): 498–505. <https://doi.org/10.1016/j.renene.2015.11.080>.

Tong, Dan, David J. Farnham, Lei Duan, Qiang Zhang, Nathan S. Lewis, Ken Caldeira, and Steven J. Davis. 2021. "Geophysical Constraints on the Reliability of Solar and Wind Power Worldwide." *Nature Communications* 12 (1): 6146. <https://doi.org/10.1038/s41467-021-26355-z>.

Wang, Mudan, Xianqiang Mao, Youkai Xing, Jianhong Lu, Peng Song, Zhengyan Liu, Zhi Guo, Kevin Tu, and Eric Zusan. 2021. "Breaking down Barriers on PV Trade Will Facilitate Global Carbon Mitigation." *Nature Communications* 12 (1): 6820. <https://doi.org/10.1038/s41467-021-26547-7>.

Xue, Yan, Carmel Margaret Lindkvist, and Alenka Temeljotov-Salaj. 2021. "Barriers and Potential Solutions to the Diffusion of Solar Photovoltaics from the Public-Private-People Partnership Perspective – Case Study of Norway." *Renewable and Sustainable Energy Reviews* 137 (March): 110636. <https://doi.org/10.1016/j.rser.2020.110636>.